# INVESTIGATING THE EFFECTIVE DIMENSIONALITY OF A MODEL USING A THERMODYNAMIC LEARNING CAPACITY

## ABSTRACT

We use a formal correspondence between thermodynamics and inference, where the number of samples can be thought of as the inverse temperature, to study a quantity called "learning capacity" which is a measure of the effective dimensionality of a model. We show that the learning capacity is a useful notion of the complexity because (a) it is a tiny fraction of the number of parameters for many deep networks trained on typical datasets and correlates well with the test loss, (b) it depends upon the number of samples used for training, (c) it is numerically consistent with notions of capacity obtained from PAC-Bayes generalization bounds, and (d) the test loss as a function of the learning capacity does not exhibit double descent. We show that the learning capacity saturates at very small and very large sample sizes; the threshold that characterizes the transition between these two regimes provides guidelines as to when one should procure more data and when one should search for a different architecture to improve performance. We show how the learning capacity can be used to provide a quantitative notion of capacity even for non-parametric models such as random forests and nearest neighbor classifiers.

## 1 INTRODUCTION

Deep networks often have many more parameters than the number of training data. Classical results in statistical learning theory suggest that such models can overfit, i.e., they can make inaccurate predictions on data outside the training set. In practice, these networks seem to defy this accepted statistical wisdom and work extremely well across a large class of problems. In fact, the test error for typical networks often improves as models with more and more parameters are fitted upon the same dataset. Resolving this paradox is one of the biggest open questions in theoretical analysis of deep learning today. §4 discusses various techniques that have been used to understand this phenomenon.

The existing body of work on this problem has found that the number of parameters is often a poor indicator of the complexity of functions that can be learned by this model on typical data, and thereby the test loss. There are a number of alternative quantities that have been proposed to rectify this, e.g., improved estimates of VC-dimension, different kinds of norms that more accurately count the actual number of parameters that make predictions (Fisher norm, spectral norm, path norm), quantities that characterize the sensitivity of the loss to perturbations of the trained weights (sharpness-based metrics, PAC-Bayes bounds), quantities that characterize the sensitivity of the solution to perturbations of the training data (stability-based bounds), etc. Jiang et al. (2019) argue that PAC-Bayes bounds most accurately reflect the test performance of typical deep networks compared to other quantities.

In this paper, we offer a different perspective of PAC-Bayes generalization bounds, and using this perspective we develop techniques to calculate a kind of "learning capacity" (akin to heat capacity in thermodynamics). The contributions of this work are as follows.

**We develop a Monte Carlo-based procedure to calculate the learning capacity of machine learning models** ranging from deep networks with different architectures (fully-connected, convolutional, residual, RNNs and LSTMs) fitted on image classification and text prediction tasks for multiple datasets, to non-parametric models such as random forests and $k$-nearest neighbor classifiers fitted to tabular data. We show that in almost all these cases, the learning capacity correlates well with the test loss, e.g., for the same sample size, models with a *smaller* learning capacity has a better test loss. This suggests that the learning capacity can be used faithfully used as a model selection criterion.

**We show that the test loss when plotted against the learning capacity does not exhibit double descent.** The test loss monotonically increases with the learning capacity for small sample sizes. For many deep networks, the learning capacity is a tiny fraction ($\sim 2\%$) of the total number of weights.

**We develop a connection between the learning capacity and PAC-Bayes generalization bounds.** We show that the learning capacity is mathematically close to a notion of effective dimensionality gleaned from PAC-Bayes theory, and also corroborate this via experiments. This makes the analogy from thermodynamics rigorous and relevant to statistical learning theory. Using singular learning theory, we also discuss the relationship of learning capacity to quantities that characterize model complexity for probabilistic models with symmetries in the parameter space in Appendix C.

**We show a "freezing" phenomenon in machine learning.** The learning capacity saturates for very small (equivalent to high temperature) and very large (equivalent to low temperature) sample sizes $N$. Models with different architectures and number of weights trained on the same dataset have the same learning capacity for small $N$—one should procure more data to improve the test loss in this regime. For large $N$, our formulae indicate a regime of diminishing returns where the test loss is proportional to $N^{-1}$ (this has also been noticed in the literature on scaling laws). In this regime, one should search for different architectures to improve the test loss—we show that in this regime, fixed the value of $N$, smaller the learning capacity, the better the test loss.

## 2 METHODS

Consider a dataset $D = \{(x_i, y_i)\}_{i=1}^N$ where $x \in \mathbb{R}^d$ denotes the input and $y \in \{1, \ldots, m\}$ denotes the ground-truth label for $m$ different categories. The probabilistic model of the labels given the input is denoted by $p_w(y \mid x)$ where $w \in \mathbb{R}^d$ are the weights, or parameters. For every weight configuration $w$, we can assign an energy

$$\hat{H}(w) = -\frac{1}{N} \sum_{i=1}^N \log p_w(y_i \mid x_i), \tag{1}$$

which is the negative log-likelihood of the training dataset given the weights $w$, e.g., the cross-entropy loss for classification. If $\varphi(w)$ denotes a prior distribution on the weight space, then the marginal likelihood, also known as the partition function in physics or the evidence in statistics, is

$$Z(N) = \int \mathrm{d}w \, \varphi(w) e^{-N\hat{H}(w)}. \tag{2}$$

This is the normalizing constant of a probability distribution $\propto \varphi(w) e^{-N\hat{H}(w)}$; under this distribution weight configurations that have a small energy $\hat{H}(w)$, i.e., a small cross-entropy loss, are more likely. LaMont & Wiggins (2019) noticed that the partition function above resembles the partition function for a constant temperature ensemble in statistical physics

$$Z(T) = \int \mathrm{d}s \, e^{-E(s)/(k_B T)}$$

where the probability of finding the system at a state $s$ is given by the Boltzmann distribution $\rho(s) \propto \exp\left(-E(s)/(k_B T)\right)$; with $k_B$ being the Boltzmann constant, $T$ the temperature and $E(s)$ the energy of a state $s$. The normalizing factor of this distribution is $Z(T)$. This suggests that we can think of the inverse temperature in statistical physics (assuming that we work in natural units with $k_B = 1$) as the number of samples:

$$\beta = T^{-1} \equiv N. \tag{3}$$

At a small sample size $N$, many weight configurations under the prior $\varphi(w)$ (i.e., many models in the hypothesis class) are consistent with the training data; this is similar to how at high temperatures many states $s$ are likely under the Boltzmann distribution. As the number of samples $N \to \infty$, fewer and fewer weight configurations can fit the training data; this is similar to how a physical system reaches the ground state with minimal energy $\min_s E(s)$ as the temperature goes to zero∗.

In thermodynamics, the partition function contains all the information about the system; many quantities of interest can be written down in terms of the partition function. For example, the average energy in physics is $U = \int \mathrm{d}s \, \rho(s) E(s) = -\partial_\beta \log Z(\beta)$, the entropy is $S = -\int \mathrm{d}s \, \rho(s) \log \rho(s) = \beta U + \log Z(\beta)$ and the heat capacity is $C = -\beta^2 \partial U / \partial \beta$ is the rate at which average energy changes with temperature $\beta^{-1}$. The correspondence between inverse temperature $\beta$ and number of samples $N$

---

∗We should emphasize that (3) is just a pattern matching exercise between elementary concepts in statistical inference and thermodynamics, both of which have been studied for more than a century. The factor of $N$ in the exponent of (2) is an algebraic manipulation of the average over the data points in (1). This analogy is neither new nor unique; see Appendix A.

allows us to interpret these quantities in the context of machine learning:

$$\text{Average Energy: } U(N) \doteq -\partial_N \log Z(N)$$

$$\text{Learning Capacity: } C(N) \doteq -N^2 \partial_N U \qquad (4)$$

$$= N^2 \partial_N^2 \log Z(N).$$

The average energy $U$ is the average loss of weight configurations sampled from the Boltzmann distribution $\varphi(w)e^{-N\hat{H}(w)}$ while the quantity $C$ which we can *define* to be "learning capacity" is the rate at which the average energy increases with temperature $N^{-1}$. It will help to bear in mind for now that average energy $\equiv$ test loss and learning capacity $\equiv$ effective dimensionality. We will make this correspondence rigorous. Roughly speaking, just like the heat capacity in thermodynamics is the amount of heat required to produce a unit change in the temperature, the learning capacity of a statistical model is proportional to the rate of drop in the test loss as more and more samples (from the same probability distribution, with standard assumptions of independence) are added to the training dataset.

It will be useful to explain why we would like to think of the learning capacity as a measure of the effective dimensionality of a model. Suppose we have a convex energy $\hat{H}(w) = 1/2 \langle w - w^*, A(w - w^*) \rangle$ with $A \in \mathbb{R}^{p \times p}$.

- If the Hessian $A \succ 0$, for any proper uniform prior, the learning capacity $C$ equals half the number of parameters $p/2$.
- If $A \succeq 0$ with $\text{rank}(A) = K$, then so long we have a proper Boltzmann distribution on the weights, e.g., $\int \mathrm{d}w\, \varphi(w)\mathbf{1}\{Aw = 0\} < \infty$, the learning capacity $C$ equals half the effective number of parameters $K/2$. This suggests that the learning capacity could be effective in discounting the symmetries in neural architectures because it ignores the combinations of parameters that lie in the null space of the Hessian.
- If the eigenvalues of the Hessian are $\lambda_i(A)$ and we have a Gaussian prior $\varphi(w) = \mathrm{N}(w^*, \epsilon^{-1}I)$, then the learning capacity is

$$C = \frac{p}{2} - \epsilon\, \text{HM}^{-1}(\lambda_i(A)) \qquad (5)$$

where $\text{HM}(\lambda_i(A))$ is the harmonic mean of the eigenvalues of $A*$. For example, if eigenvalues $\lambda_i(A) = 1/i$, then $\text{HM}^{-1}(\lambda_i(A)) \approx -N^{-1}\log(\lambda_{\max}/\lambda_{\min})$. If the eigenvalues of $A$ are small (large $\text{HM}^{-1}$) or if the condition number of Hessian $A$ is large then the learning capacity can be much smaller than total number of parameters—in both cases, there exist directions in the weight space such that perturbations along them do not affect the loss much.

In the coming sections, we will make such statements more precise and connect the concept of learning capacity to PAC-Bayes theory.

## 2.1 Estimating the average energy $\overline{U}$ and learning capacity $\overline{C}$

The Bayesian predictive log-likelihood of a new datum $x$ is†

$$\log p(y \mid x, D) = \log \frac{\int \mathrm{d}w\, p_w(y \mid x)\varphi(w)e^{-N\hat{H}(w)}}{\int \mathrm{d}w\, \varphi(w)e^{-N\hat{H}(w)}} = \log Z(N+1) - \log Z(N) \approx -U, \qquad (6)$$

where in the first step we sent the likelihood of the new datum $p_w(y \mid x)$ into the exponent. The final approximation is reasonable because unlike statistical physics where the inverse temperature is a scalar, our inverse temperature $N$ takes integer values. This shows that the average test likelihood of a datum (up to a constant that is the entropy of the true labels) is

$$- \mathop{\mathbb{E}}_{x,y \sim P} [\log p(y \mid x, D)] \approx \overline{U} = \overline{\log Z}(N+1) - \overline{\log Z}(N); \qquad (7)$$

where $\overline{f} = \mathbb{E}_{x,y \sim P}[f(x,y)]$ for any function $f$. This is an important identity: the rate of increase of the log-partition function with respect to the number of samples $N$ in the training dataset $D$ is equal to the average test negative log-likelihood. The leave-one-out cross-validation estimator (LOOCV) allows us to estimate the average over the probability distribution $P$, i.e.,

$$\overline{U} = -\partial_N \overline{\log Z}(N) \approx \overline{\log Z}(N-1) - \overline{\log Z}(N) = -N^{-1}\sum_{i=1}^{N} \log p(y_i \mid x_i, D^{(-i)}), \qquad (8)$$

---

∗This calculation uses a perturbation approximation of the eigenvalues: $\lambda_i(A + \epsilon I/N) \approx \lambda_i(A) + \epsilon/N$.

†We introduce the setup for classification problems but these expressions can also be written down for regression: if targets $y_i \in \mathbb{R}$ and if $f_w(x)$ is the prediction on the input $x$, we use $\log p_w(y \mid x) = -\frac{(y - f_w(x))^2}{2\sigma^2} + \text{constant}$, for some non-zero standard-deviation $\sigma$ to calculate the energy in (1).

where $D^{(-i)}$ denotes a training dataset of $(N-1)$ samples with the $i^{\text{th}}$ sample removed. Given an estimate for the average energy $\overline{U}(N)$ for different values of $N$, we will estimate the learning capacity $\overline{C}$ by taking its derivative.

Estimating the test loss using the LOOCV estimator will require us to fit $N$ different models to calculate each of the terms in the summation above. In practice, we do so for large datasets and large models using the $k$-fold cross-validation.

## 2.2 Numerical estimates of the average energy and learning capacity using Monte Carlo

For each model class and dataset, e.g., different deep networks, random forests and different datasets, we perform the following steps to first estimate the average energy (averaged over multiple sample sets) $\overline{U}(N)$. For each value of $N$, sub-sample $n$ different datasets $D_N^{(i)} \subseteq D$ for $i \in \{1, \ldots, n\}$ with replacement (i.e., bootstrapped versions of the datasets), each with $N$ samples. Create $k$-folds within each dataset $D_N^{(i)}$ and for each fold. Let the $j^{\text{th}}$ fold be $\xi^{(j)} \subset D_N^{(i)}$; this has $(1 - 1/k)N$ samples in the training set and $N/k$ samples in the test set. For each fold, train $m$ models, say denoted by parameters $w^{(l)}; l \in \{1, \ldots, m\}$ from different initializations (random seeds in practice) and calculate

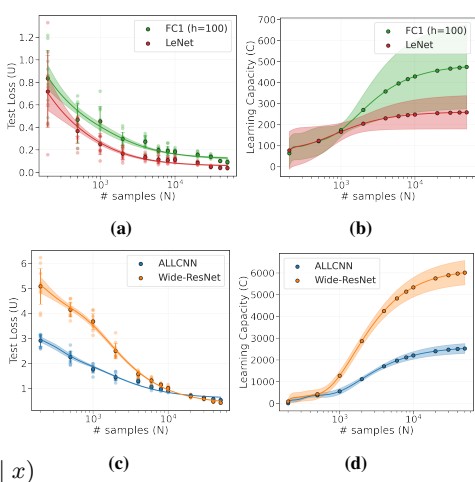

**(a)**          **(b)**

**(c)**          **(d)**

$$\overline{U}(N) = -\frac{1}{nmN} \sum_{i=1}^{n} \sum_{j=1}^{k} \sum_{(x,y) \in \xi^{(j)} \subset D_N^{(i)}} \sum_{l=1}^{m} \log p_{w^{(l)}}(y \mid x) \tag{9}$$

For all experiments, we set the the number of bootstrapped datasets $n = 4$, the number of folds $k = 5$, and the number of different models trained on each fold $m = 5$. For each value of $N$ (we roughly use 15 values; close by at small sample sizes and far apart for large sample sizes), this amounts to training 100 models to estimate $\overline{U}$.

**Figure 1:** **(a,c)** show the average energy ($\overline{U}(N)$) for one-hidden-layer fully-connected network with 100 neurons and LeNet on MNIST (binary classification) and the ALLCNN and Wide-ResNet on CIFAR-10 (10-class classification in this case). **(b,d)** shows the learning capacity ($\overline{C}(N)$) estimated by fitting a seventh-order polynomial to $\overline{U}$ with constraints on it being monotonically decreasing and a constraint on $\overline{C}$ increasing monotonically.

Given these estimates of the average energy, we could now calculate $\overline{C} = -N^2 \partial_N \overline{U}$ but it is difficult to do so because the noise in the estimation of $\overline{U}$ gets amplified when we take the finite-difference derivative. In preliminary experiments, we used a polynomial model to fit $\overline{U}(N)$ (with the constraint that it decreases monotonically with $N$ since it is the test loss) and calculated the analytical derivative of this polynomial to obtain $\overline{C}(N)$; this procedure also gives an estimate of the uncertainty on both $\overline{U}$ and $\overline{C}$. This works reasonably well (Fig. 1).

But we also noticed that the learning capacity $\overline{C}(N)$ typically looks like a sigmoid, i.e., it saturates at both very small and very large sample sizes with a sharp rise at some intermediate value. We therefore fit a sigmoid

$$\overline{C}(N) = \frac{a}{1 + \exp(-c \log N + b)} \tag{10}$$

with coefficients $a, b, c$ and the independent variable $N$. One may think of the coefficient $a$ as the maximal learning capacity of the model with $N \to \infty$. We analytically integrate this expression to get $\overline{U}(N) = -\int_1^N dk \, k^{-2} \overline{C}(k)$ in terms of the unknown coefficients. We fit this expression to the data using the Levenberg-Marquardt algorithm and obtained accurate fits with small uncertainties. Using the sigmoid circumvents the need to choose sensitive hyper-parameters such as the degree of the polynomial. We found that both polynomial regression and this sigmoid-based technique give similar estimates of the learning capacity for all the experiments in this paper. We have also developed an alternative way to estimate $\overline{U}$ and $\overline{C}$ using Markov chain Monte Carlo in Appendix B which obtains comparable results.

## 2.3 Interpreting the learning capacity in the PAC-Bayesian framework

The PAC-Bayesian framework (Langford & Seeger, 2001; McAllester, 1999) estimates the population error of a randomized hypothesis with a distribution $Q$ using its empirical error and its KL-divergence with respect to some prior $\varphi$. For any $\delta > 0$, with probability at least $1 - \delta$ over draws of the samples

in the training dataset, we have

$$e(Q) \leq \hat{e}(Q) + \frac{\text{KL}(Q,\varphi)}{N} \tag{11}$$

where $e(Q)$ is the average population error of hypotheses sampled from $Q$ and $\hat{e}(Q)$ is the analogously defined training error. We have purposely used a loose version of the bound (the usual one has a square root on the complexity term) because we will need it soon. PAC-Bayes theory has been fruitfully used to calculate non-vacuous generalization bounds, even for deep networks (Dziugaite & Roy, 2017; Wu et al., 2021). Jiang et al. (2019) have also argued that PAC-Bayes generalization bounds correlate better with test error for deep networks than other quantities used to understand generalization.

For a Gaussian prior $\varphi = \text{N}(w_0, \epsilon^{-1}I)$ (which is a user-chosen parameter) and a Gaussian posterior $Q = \text{N}(w, \Sigma_q)$ centered at a fixed weight configuration $w$ (usually taken to be the weights obtained at the end of the training process), Yang et al. (2022b) calculated the above version of the PAC-Bayes bound analytically:

$$e(Q) \leq \frac{\sum_{i=1}^{p(N,\epsilon)} \log\left(2(N-1)\lambda_i + \epsilon\right) + 2/\kappa + \epsilon\|w - w_0\|_2^2}{4(N-1)} \tag{12}$$

where $\kappa$ is the largest constant such that the ratio of the eigenvalues (sorted by magnitude) of the Hessian of the loss at $w$ satisfies $\log \lambda_i/\lambda_r = -\kappa(i-r)$ beyond a threshold number of dimensions

$$r = p(N,\epsilon) \doteq \sum_{i=1}^{p} \mathbf{1}\left\{|\lambda_i| \geq \frac{\epsilon}{2(N-1)}\right\}. \tag{13}$$

In other words, if the eigenvalues of the Hessian at $w$ decay quickly (large $\kappa$) after $p(N,\epsilon)$ dimensions, then the PAC-Bayes bound is small. They also showed that the analytical expression in (12) is a non-vacuous bound for fully-connected and LeNet architectures on MNIST. Effectively, the PAC-Bayes posterior spreads its probability mass away from $w$ in a subspace spanned by the small eigenvectors of the Hessian—only $p(N,\epsilon)$ combinations of weights are strongly constrained by the training data. The bound in (12) can be shown to be equal to

$$\mathcal{O}(p(N,\epsilon)/N + \sqrt{p(N,\epsilon)/N}). \tag{14}$$

And therefore, they proposed $p(N,\epsilon)$ as the "effective dimensionality" of a model.

For general non-Gaussian priors and posterior distributions, we can substitute $e(Q) = \int \mathrm{d}w\, Q(w)\hat{H}(w)$ in (11) and simplify to get

$$\min_Q \hat{e}(Q) + \frac{\text{KL}(Q,\varphi)}{N} = \min_Q \frac{\text{KL}(Q, p(w \mid D))}{N} - \frac{\log Z(N)}{N},$$

where $Z(N) = \int \mathrm{d}w\, \varphi(w)e^{-N\hat{H}(w)}$ like we have defined in (6) and the minimum is achieved when $Q = p(w \mid D)$. Therefore the looser version of the PAC-Bayes bound above is effectively calculating (after taking a second order Taylor series approximation of the energy $\hat{H}(w)$, and using a perturbation approximation of the eigenvalues of the posterior covariance)

$$\begin{aligned}
-\frac{\log Z(N)}{N} &\approx \frac{p \log N}{2N} + \frac{\sum_i \log \lambda_i}{2N} + \sum_i \frac{\epsilon}{2N\lambda_i} + \frac{\epsilon\|w^* - w_0\|^2}{N} \\
&= \frac{p + \epsilon\,\text{HM}^{-1}(\lambda_i)}{2N} + \frac{\sum_i \log \lambda_i}{2N} + \frac{\epsilon\|w^* - w_0\|^2}{N} \\
&= \frac{C}{N} + \frac{\epsilon\|w^* - w_0\|^2}{N}, \quad \text{from (5),}
\end{aligned} \tag{15}$$

for $\epsilon = -2\text{HM}(\lambda_i)\left(\sum_i \log \lambda_i\right)$ where we have used the short-form $\lambda_i \equiv \lambda_i(\nabla^2 \hat{H}(w^*))$ for the eigenvalues of the Hessian of the loss at $w^*$∗. In other words, the coefficient in the PAC-Bayes bound in front of $1/N$ is equal to the learning capacity $C = -N^2 \partial_N U$ up to a constant is proportional to the distance of the solution from initialization. Our techniques to estimate the learning capacity are computationally inexpensive ways to calculate PAC-Bayes bounds.

## 3 RESULTS

We conducted a comprehensive investigation using a diverse range of models. These models include traditional models (random forests and $k$-nearest neighbor classifiers) and different deep learning ar-

---

∗It may not seem appropriate to select a value of $\epsilon$ using the eigenvalues of the Hessian of the loss, but we can imagine (like it is done for data-distribution depending PAC-Bayes calculations) that these eigenvalues are estimated using some held-out samples.

| Architecture | Dataset | Test Accuracy (%) | Test Loss ($\overline{U}(N)$) | Weights ($p$) | Learning Capacity ($\overline{C}(N)$) | $\overline{C}(N)/p$ (%) | Kendall's $\tau$ coefficient |
|---|---|---|---|---|---|---|---|
| FC-1-100 | MNIST | $97.6 \pm 0.1$ | $0.09 \pm 0.01$ | 78,902 | $609 \pm 560$ | 0.8 | 0.40 |
| LeNet | | $98.9 \pm 0.1$ | $0.04 \pm 0.01$ | 43,746 | $231 \pm 204$ | 0.5 | |
| ALLCNN | CIFAR-10 | $88.9 \pm 0.3$ | $0.53 \pm 0.02$ | 262,238 | $4571 \pm 1700$ | 1.7 | 0.22 |
| Wide-ResNet | | $93.7 \pm 0.1$ | $0.43 \pm 0.02$ | 298,542 | $5805 \pm 1145$ | 1.9 | |
| RNN | WikiText-2 | - | $5.53 \pm 0.02$ | 13,480,817 | $5777 \pm 1151$ | 0.04 | -0.03 |
| LSTM | | - | $4.86 \pm 0.01$ | 13,962,014 | $14762 \pm 2345$ | 0.1 | |
| | **Synthetic ($\kappa$)** | | | | | | |
| | 0.1 | $93.7 \pm 0.2$ | $0.20 \pm 0.01$ | | $1117 \pm 221$ | 1.4 | 0.15 |
| FC-1-100 | 1 | $93.9 \pm 0.2$ | $0.19 \pm 0.01$ | 78,902 | $870 \pm 338$ | 1.1 | -0.00 |
| | 10 | $98.1 \pm 0.3$ | $0.05 \pm 0.00$ | | $112 \pm 39$ | 0.1 | 0.10 |
| | 20 | $98.8 \pm 0.3$ | $0.05 \pm 0.00$ | | $70 \pm 19$ | 0.1 | 0.09 |

**Table 1:** The learning capacity $\overline{C}(N)$ is a tiny fraction of the total number of weights in the network for a wide variety of architectures on different datasets $N = 2M$ for WikiText-2 and $N = 50K$ for all others. We evaluated the utility of using the learning capacity as a model selection criterion by calculating Kendall's $\tau$ coefficient across different models at a fixed sample size $N$ (the rightmost column reports the average over $N$). For all architectures on MNIST and CIFAR-10, learning capacity correlates well with the test loss; correlation is poor for WikiText-2. We also calculated an "intra-architecture" Kendall's $\tau$ coefficient (not shown in the table) where for each dataset, we check the correlation of $\overline{C}$ and $\overline{U}$ over the bootstrapped datasets, different sample sizes, and different architectures. This coefficient is more negative than -0.79 for all architectures which indicates that the learning capacity is insensitive to these hyper-parameters (it is negative because if the architecture is held fixed, higher the capacity smaller the test loss). Another pattern to note here is that larger the difficulty of the dataset, larger the learning capacity (e.g., CIFAR-10 vs. MNIST, synthetic datasets with a small $\kappa$ are harder). See Appendix D for details.

chitectures (multi-layer perceptrons, convolutional networks including LeNet, AllCNN (Springenberg et al., 2015), residual and wide-residual networks, recurrent models and LSTMs). These models are trained under various settings using datasets such as MNIST, CIFAR-10, WikiText-2, multiple tabular datasets and synthetic datasets. Appendix D gives more details∗.

## 3.1 LEARNING CAPACITY FOR DEEP NETWORKS

Table 1 and Fig. 2 show the learning capacity for different deep learning architectures trained on various datasets and a comparison with the number of weights in each architecture. **In all cases, the learning capacity $\overline{C}$ at the maximum sample size is a tiny fraction of the number of weights.** This observation is consistent many results in deep learning, e.g., pruning (Molchanov et al., 2019), lottery ticket hypothesis (Frankle & Carbin, 2019), distillation (Hinton et al., 2015), and also with many results in physics and systems biology (Transtrum et al., 2011). This shows that a surprisingly few degrees of freedom are constrained by the data in these models even for large sample sizes.

**The networks in Fig. 2 exhibit a freezing phenomenon where the learning capacity saturates at large and small sample sizes** (except AllCNN and LSTM at large $N$). Different architectures have the same capacity at small sample sizes; this suggests that when the amount of information in the training dataset is small, a small fraction of the degrees of freedom is constrained. Different architectures have different capacities at large $N$ for the same dataset. **But fixed a sample size, even across different architectures smaller the capacity the better the test loss (see Table 1, this trend does not hold for WikiText-2).** And therefore, we can fruitfully use the learning capacity for model selection among different models trained on the same dataset. From the expressions in (4), we can see that $\overline{U}$ decreases as $\approx 1/N$ if $\overline{C}$ is a constant. This suggests that we should search for different architectures in the saturation regime when there are diminishing returns from increasing $N$ if the sample size is already large.

We studied the test loss and learning capacity for fully-connected student networks fitted on **synthetic datasets of varying degrees of complexity** labeled by a fixed random teacher in Fig. 2c. Inputs of these datasets are sampled from a Gaussian with zero mean and a covariance matrix whose eigenvalues decay exponentially with different slopes; larger the slope, smaller the effective rank of the dataset and more (effectively) low-dimensional the true model. In Table 1, the learning capacity

---

∗These experiments required a lot of Monte Carlo and Markov Chain Monte Carlo runs. We present data from about 35,000 networks of varying sizes and complexities, about 10,000 random forest and $k$-NN-based classifiers were trained across multiple datasets. We also optimized a data-distribution dependent PAC-Bayes bound using the approach of Dziugaite & Roy (2018); Yang et al. (2022b) and calculated the eigenvalues of the Hessian for 2,800 models, both of which are very expensive computationally.

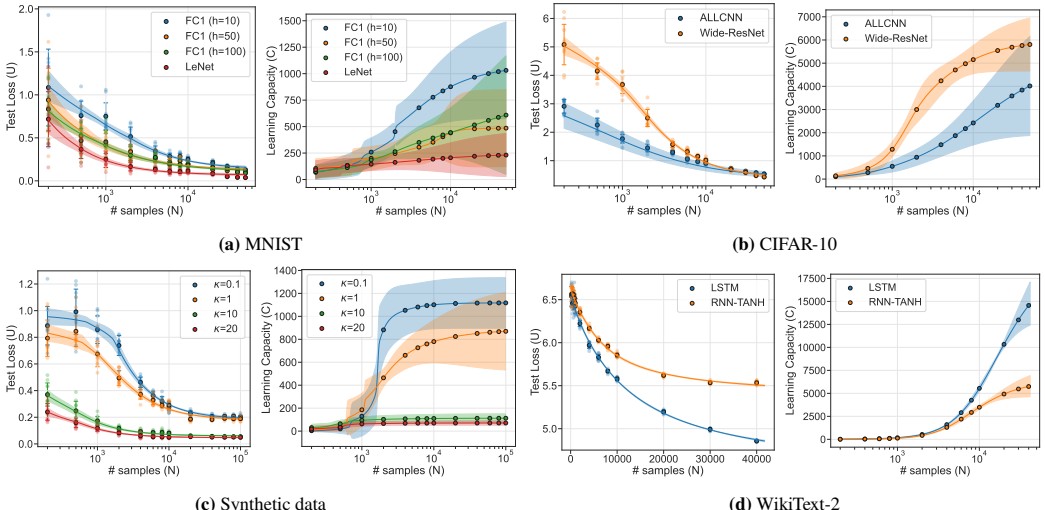

**(a)** MNIST

**(b)** CIFAR-10

**(c)** Synthetic data

**(d)** WikiText-2

**Figure 2:** Average energy $\overline{U}(N)$ and learning capacity $\overline{C}(N)$ for different architectures, datasets and input modalities. Table 1 shows the rightmost point on these plots. The average energy decreases monotonically as a function of the number of samples $N$ in the training dataset. The learning capacity, which indicates the number of constrained degrees of freedom in a model, increases with the number of training samples $N$. Architectures with a smaller learning capacity have a smaller test loss. Different architectures trained on the same dataset have different learning capacity. The learning capacity exhibits freezing at both very small and very large $N$ for many of these models. For synthetic data experiments in (c), the learning capacity of models with the same architecture is smaller if the task is easier (large values of $\kappa$).

accurately reflects the difficulty of the task; the same architecture when trained on an easier task has a smaller learning capacity. This trend is perfectly consistent with the experimental results using PAC-Bayes generalization bounds on similar datasets by Yang et al. (2022b) who also find that simpler the task, smaller the effective dimensionality $p(N, \epsilon)$ in (13) and that the same architecture uses more degrees of freedom to make predictions on difficult tasks.

Susceptibilities in thermodynamics (e.g., heat capacity or bulk modulus) quantify the changes in an extensive property (e.g., the average energy) under variations of an intensive property (e.g., the temperature) and are very useful to systematically characterize materials. Our results show that **some architectures are better suited to some datasets than others**, e.g., even if the wide residual network is both bigger and regarded to be "better" than AllCNN, the latter may be better suited for some problems in terms of how quickly the test loss decreases with the number of samples. While the test loss/error of all the models on MNIST is quite similar at large sample sizes, the convolutional network is using very few degrees of freedom to make accurate predictions.

### 3.2 TEST LOSS AS A FUNCTION OF THE LEARNING CAPACITY DOES NOT EXHIBIT DOUBLE DESCENT

Fig. 4 shows the test loss as a function of the learning capacity for a large variety of networks (fully-connected networks with number of hidden neurons ranging from 10 to 100 trained on MNIST using samples ranging from $N = 200$ to $N = 50,000$ each shown in a different color) and contrasts this with the double descent curve for $N = 50,000$. In all cases, the slope of a regression fitted to the test loss (equivalent to average energy $\overline{U}$) as a function of the learning capacity $\overline{C}$ is positive (except for $N = 50,000$ where the $p$-value for the slope being non-zero is not significant). This is a strong validation of the fact that learning capacity can adequately measure the complexity of a learned model. Indeed a good metric of complexity should be strongly correlated with the test loss. **This is a reassurance that even in deep learning, we should select models with a smaller complexity to obtain a small test loss—not necessarily a small number of weights but a small learning capacity.**

The curve of test loss versus learning capacity is monotonic, as opposed to a U-shaped curve in the bias-variance tradeoff. We expect from statistical physics calculations (LaMont & Wiggins, 2019) that

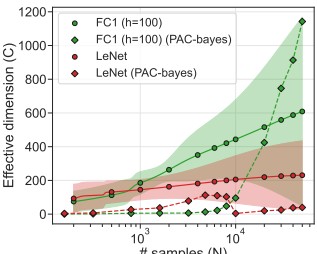

**Figure 3:** Learning capacity $\overline{C}$ (solid lines) is numerically consistent with the effective dimensionality obtained from a PAC-Bayes bound in (13) (dotted lines). Kendall rank correlation between the learning capacity and the PAC-Bayes effective dimensionality is 0.99 ($p$ = 4E-9) for the fully connected network and 0.2 ($p$ = 0.37) for LeNet.

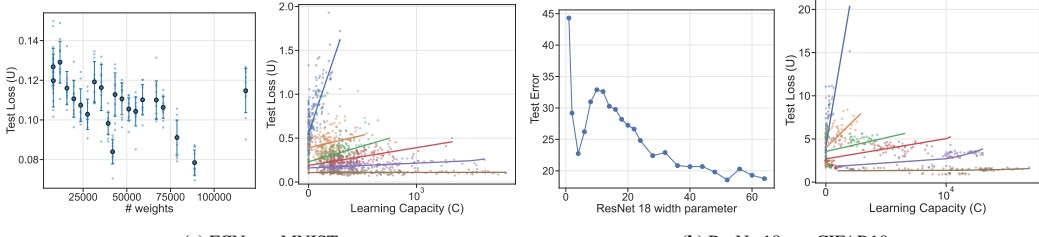

(a) FCNs on MNIST          (b) ResNet18s on CIFAR10

**Figure 4: (a) Left:** double descent phenomenon for the test loss as a function of the number of weights for one and two-layer fully-connected networks with different numbers of hidden neurons (10–100) trained on MNIST with sample size $N$ = 50,000. **(a) Right:** double descent phenomenon for ResNet18 with different width (1-64) trained on CIFAR10 with sample size $N$ = 50,000 and with noise rate 0.2. **(b):** when plotted against the learning capacity the test loss does not exhibit double descent. Colors indicate different values of $N$ (blue: 200, orange: 1000, green: 2000, red: 4000, purple: 10000, and brown: 50,000). Slope of all regressions is non-zero and positive ($p < 0.005$) except the brown curve for which the $p$-value is not significant.

the test loss (average energy) should be monotonic in the learning capacity (heat capacity) so this is not surprising *per se*. But it does seem unusual that even with our new way of measuring the capacity, we still do not obtain the classical U-shaped curve. We believe this is because the learning capacity estimates the capacity not of the entire model class, but a subset thereof—we suspect that this is the set of models that are reachable using stochastic optimization on samples from the training dataset. Mao et al. (2022) have argued that the set of such reachable models is a very small fraction of the set of all models. Effectively, our Monte Carlo based procedure to calculate learning capacity is only integrating the log-partition function within this small set.

### 3.3 COMPARING LEARNING CAPACITY WITH PAC-BAYES-BASED EFFECTIVE DIMENSIONALITY

We compute the effective dimensionality in (13) using eigenvalues of the Hessian of trained network computed using a Kronecker-decomposition of the block-diagonal Hessian using Backpack (Dangel et al., 2020). We then optimize the posterior of the (square root version) PAC-Bayes bound numerically to compute the best scale $\epsilon$ for the prior $\varphi = \mathrm{N}(w_0, \epsilon^{-1}I)$. See Dziugaite & Roy (2017) for details. As Fig. 3 shows, **our estimates for learning capacity are of the same order as the effective dimensionality estimated from PAC-Bayes theory**—and both are much smaller than the number of weights in the network. This lends credence to our definition of the learning capacity.

The learning capacity and the PAC-Bayes-based estimate of the effective dimensionality were calculated in very different ways, e.g., the former trained models independently on different folds of data whereas the latter optimized the posterior $Q$ centered at a trained network; the former did not optimize the prior from which weights were sampled while beginning training and the latter explicitly optimized the prior to get a tighter bound, etc. The fact that the two quantities are consistent with each other therefore also validates our procedure to compute the learning capacity. **Optimizing PAC-Bayes bounds numerically is very challenging technically and computationally**. It has been difficult to obtain non-vacuous bounds on datasets more complicated than MNIST. In contrast, the **learning capacity is much more straight-forward to compute**.

### 3.4 LEARNING CAPACITY FOR NON-PARAMETRIC MODELS

Our technique used to estimate the learning capacity is general and does not require explicit representation of the parameters of the model. On three tabular datasets, Table 1 and Fig. 5 shows the estimates of the learning capacity for non-parametric models such as $k$-nearest neighbor classifiers and random forests. These estimates correlate well with the test loss. We observe freezing at small sample sizes where the learning capacity is the same for all datasets but do not see freezing at large sample sizes. For random forests, we can easily have situations when the number of leaves is as large as $10^7$. The VC-dimension of a decision tree is proportional to the number of leaves (Leboeuf et al., 2020), so such models are also highly over-parameterized when they are fitted on fewer samples than the number of leaves. As we show in Fig. S.2, the learning capacity is typically much smaller than the number of leaves irrespective of the dataset—by about 3 orders of magnitude. This is a remarkable finding because models such as random forests and $k$-NNs do not have a well-defined objective that is optimized; due to this the PAC-Bayes-based estimation of the effective dimensionality or the log-canonical threshold cannot be defined for these models.

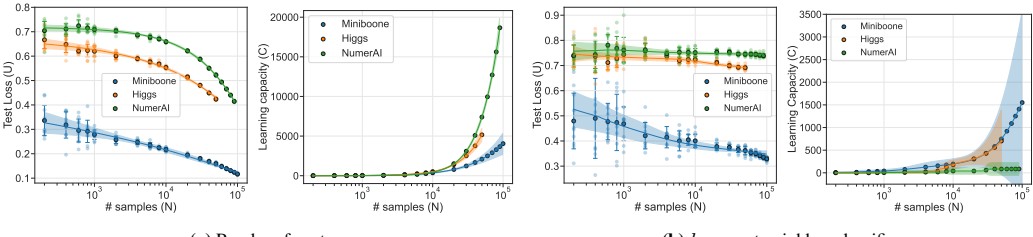

**(a)** Random forest                       **(b)** $k$-nearest neighbor classifier

**Figure 5:** Average energy $\overline{U}(N)$ and learning capacity $\overline{C}(N)$ for three tabular learning problems using random forests and $k$-nearest neighbor classifiers. In both cases, the Miniboone dataset has the best test loss and both models exhibit a lower learning capacity for this dataset (for the $k$-NN there is a large uncertainty in the estimate in this case). As the random forest learns the Numerai dataset, its capacity (which is reflective of the number of constrained degrees of freedom) increases sharply; in contrast the $k$-NN does not have a good test loss on this dataset and its capacity is much smaller.

## 4    DISCUSSION AND RELATED WORK

**Learning theory** Although there is extensive work on model selection in both information theory, e.g., (Rissanen, 1978) and machine learning (Friedman et al., 2001), performing effective model selection for deep networks has proven difficult (Zhang et al., 2017). The key issue is that the number of weights in a deep network do not directly relate to the complexity of the learned model (Bartlett et al., 2017). Many quantities have been proposed to fix this, e.g., to estimate the complexity of the hypothesis class (Liang et al., 2019; Neyshabur et al., 2016; Poggio et al., 2020), or that of the learned model (Jiang et al., 2019; Shwartz-Ziv & Tishby, 2017; Achille & Soatto, 2019; Chaudhari et al., 2017). There are many broad similarities between these approaches and our thermodynamics-based approach, e.g., compression bounds perturb input using noise, flatness-based metrics perturb the weights, while the learning capacity perturbs the number of samples; we also discuss the connection to PAC-Bayes theory in §2.3 and singular learning theory in Appendix C.

    **Double descent phenomenon** There is an extensive body of theoretical work that studies the double descent phenomenon (Belkin et al., 2019), e.g., using two-layer networks (Mei & Montanari, 2022; Mei et al., 2022), linear models (Hastie et al., 2019). The double descent phenomenon is surprising for two reasons: the sharp increases in the variance of the estimator near the interpolation threshold and the monotonic decrease in the test error as the number of parameters in the model grows. The former can be eliminated by regularization (Nakkiran et al., 2020b) but to our knowledge this paper presents the first demonstration that the latter can also be addressed. Our results show that at small sample sizes, when the test loss is plotted against the learning capacity the test loss grows with increasing capacity—this is akin to the later half of the classic U-shaped curve in the bias-variance tradeoff. For large sample sizes, the test loss saturates as a function of the learning capacity—this indicates that the learning capacity adequately captures the effective dimensionality of the model and even if the number of bare parameters in the hypothesis class grows, the number of degrees of freedom that are constrained by the data does not grow.

    **Effective dimensionality from replica theory calculations in statistical physics** For perceptron-like models, calculations in statistical physics (Engel & den Broeck, 2001; Watanabe, 2009; Baldassi et al., 2015) show that the generalization error (akin to our $\overline{U}$) is inversely proportional to the sample size $N$, e.g., $0.625d/N$ and $0.442d/N$ for the so-called Gibbs and Bayes learning rules respectively. The "effective dimensionality" of these rules can be calculated using (4) also saturates at large $N$. Such insensitivity of the performance to the number of samples has also been noticed in the so-called "deep bootstrap" phenomenon (Nakkiran et al., 2020a); this has also been studied theoretically using kernel gradient flows (Ghosh et al., 2021; Simon et al., 2021). Our results show a more refined phenomenon: the learning capacity, which is a good estimate of the test loss, saturates at both small and high sample sizes and a sharp transition in between.

    **Freezing phenomena in disordered systems** The equipartition theorem says that a free particle with $p$ degrees of freedom confined to a volume has a heat capacity of $p/2$. As LaMont & Wiggins (2019) discuss, in many physical systems, the heat capacity can be smaller. Such deviation from the equipartition theorem is well-known. At very high temperatures, the degrees of freedom in a system can become irrelevant and therefore stop contributing to the heat capacity. Similarly at very low temperatures, there may not be sufficient energy to populate all energy levels and this again leads to a deviation. In physical systems, such freezing can also occur at intermediate temperatures.

    **See Appendix C and Appendix G for more discussion points.**

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

# A    DIFFERENT ANALOGIES BETWEEN THERMODYNAMICS AND STATISTICAL INFERENCE

There have been many other attempts over the years to explore the conceptual similarities between thermodynamics and Bayesian inference. The analogy between thermodynamics and statistical inference is not unique. Depending upon our goal, it can take different forms that are unrelated to each other.

**A variational formulation of stochastic gradient descent**    A continuous-time model of stochastic gradient descent (SGD) as a Langevin dynamics with state-dependent noise is

$$\mathrm{d}w\left(t\right) = -\nabla \hat{H}(w)\,\mathrm{d}t + \left(\frac{\eta}{N_0}D(w)\right)^{1/2}\mathrm{d}B\left(t\right)$$

where $\nabla \hat{H}(w)$ is the gradient of the training objective (cross-entropy loss), $\eta > 0$ is the step-size or learning rate of SGD, $N_0 < N$ is the mini batch-size, $D(w) = \mathrm{Cov}(\nabla \hat{H}(w))$ is the covariance of the mini-batch gradient and $B(t)$ is standard Brownian motion (Chaudhari & Soatto, 2018). We can think of the factor $\eta/N_0$ as a temperature; larger the learning rate larger the effective noise in the SGD weight updates as compared to gradient descent, and larger the mini-batch size $N_0$ smaller the discrepancies between the SGD gradient and the full gradient. For this model, with some technical assumptions, the probability density over the weight space $\rho^*$ induced by SGD is

$$\rho^*(w) = \operatorname*{argmin}_{\rho} \mathcal{F}(\rho) := \left\{ \mathbb{E}_{w\sim\rho}\left[\Phi(w)\right] - \frac{\eta}{N_0}H(\rho) \right\}$$

where $H(\rho) = -\int \mathrm{d}w\,\rho(w)\log\rho(w)$ is the Shannon entropy of the density $\rho$ and the first term is like an average energy where the function $\Phi(w)$ gets minimized on average over draws of weights from the probability density $\rho^*$. For this setup, the free energy is $\mathcal{F}(\rho)$, the temperature is $\eta/N_0$ and the updates of SGD can be understood as performing Bayesian inference using a uniform prior over the weight space, i.e., where the Kullback-Leibler divergence $\mathrm{KL}(\rho,\varphi) = H(\rho)$ when $\varphi$ is uniform, say on a compact set. The analogy above is useful to understand the steady-state distribution of SGD using ideas from non-equilibrium thermodynamics.

**A variant of Bayes law in singular learning theory**    Yet another variant is found in Watanabe's work which uses a different definition of Bayesian inference in his development of singular learning theory (Watanabe, 2009). He studies situations when the likelihood term in Bayes law $p_w(y \mid x)$ is modified to be $p_w^\beta(y \mid x)$ for an arbitrary power $\beta$. This analogy is useful for the purposes of numerically calculating Bayesian posteriors, e.g., using Markov chain Monte Carlo methods, because the posterior is continuous in this new inverse temperature $\beta$. But this changes the definitions of the average energy and it no longer corresponds to the test negative log-likelihood.

# B    ESTIMATING THE LEARNING CAPACITY USING MARKOV CHAIN MONTE CARLO

The procedure discussed above to estimate the learning capacity is computationally expensive because it requires us to fit many models with different weight initializations, on different sample sets for each sample size $N$ to calculate the average energy $\overline{U}$. We also investigated a different technique to estimate the learning capacity using Markov Chain Monte Carlo (MCMC) methods that we describe next. The posterior distribution on the weights can be written as

$$p(w; N) = \frac{1}{Z(N)}e^{-N\hat{H}(w)}\varphi(w).$$

using (1) and (2). Observe that

$$
\begin{aligned}
U(N) &= -\partial_N \log Z(N) \\
&\approx \frac{1}{Z(N)}\left(Z(N) - Z(N+1)\right) \\
&= \frac{1}{Z(N)}\int \mathrm{d}w \left(e^{-N\hat{H}(w;N)} - e^{-(N+1)\hat{H}(w;N+1)}\right)\varphi(w) \\
&= \frac{1}{Z(N)}\int \mathrm{d}w\, e^{-N\hat{H}(w;N)}\left(1 - e^{-(N+1)\hat{H}(w;N+1)+N\hat{H}(w;N)}\right)\varphi(w) \qquad (16) \\
&= \int \mathrm{d}w\,\varphi(w)\frac{e^{-N\hat{H}(w;N)}}{Z(N)}\left(1 - p_w(y_{N+1} \mid x_{N+1})\right) \\
&\approx \frac{1}{m}\sum_{i=1}^{m}\left(1 - p_{w^{(i)}}(y_{N+1} \mid x_{N+1})\right)
\end{aligned}
$$

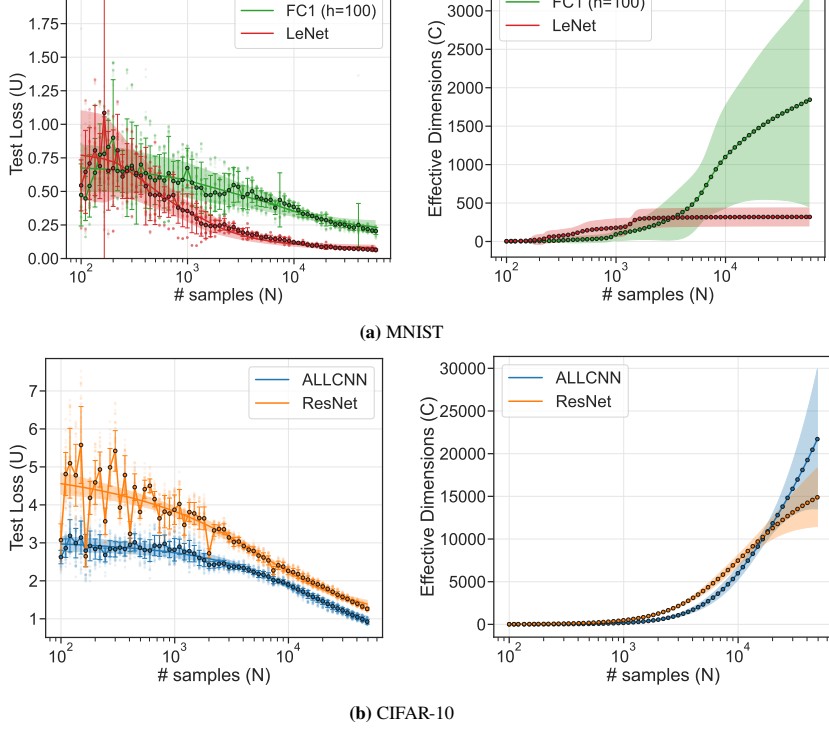

**(a)** MNIST

**(b)** CIFAR-10

**Figure S.1:** **(a,c)** show the estimate of the average energy $\overline{U}(N)$ for different architectures, for MNIST and CIFAR-10 datasets respectively. The numerical estimates of the average energy are higher than those in Fig. 2 and we also see marked jitter on these curves. This noise in the derivative translates to much higher estimates of the learning capacity $\overline{C}$ in **(b,d)** than those in Fig. 2. Even in this case, some of the broad patterns are preserved, e.g., the learning capacity is a tiny fraction of the number of weights, it saturates at both small and high sample sizes; there is no low-temperature saturation for CIFAR-10.

where $w^{(i)}$ are samples from a Markov chain that draws samples from $p(w; N)$ and that is why we can approximate the integral using the $m$ samples. We can also calculate an average version of this expression to get

$$\overline{U}(N) = \frac{1}{m} \sum_{i=1}^{m} \underset{(x,y)}{\mathbb{E}} \left[ 1 - p_{w^{(i)}}(y \mid x) \right].$$

This expression is an alternative to (8) with the key difference that we are drawing samples from the posterior $p(w; N)$ using a Markov chain instead of the leave-one-out cross-validation procedure in (8). This new expression is computationally more convenient because we only need to run one MCMC chain to draw all the samples. We implement the MCMC chain using stochastic gradient Langevin dynamics (SGLD) (Welling & Teh, 2011).

The learning capacity $C = -N^2 \partial_N U$ depends on the derivative of $U$ which allows us to calculate

$$-N^{-2}\overline{C} = \overline{\partial_N U} = \overline{U}(N+1) - \overline{U}(N)$$

$$\approx \frac{1}{m} \sum_{i=1}^{m} \underset{(x,y)}{\mathbb{E}} \left[ p_{w_N^{(i)}}(y \mid x) - p_{w_{N+1}^{(i)}}(y \mid x) \right].$$

In principle, we should run two Markov chains, one that draws samples $w_N^{(i)}$ and another, completely independent one, that draws samples $w_{N+1}^{(i)}$. Since we need to estimate $\overline{C}(N)$ for many different values of the sample size $N$, this would again require us to implement many different Markov chains. In practice, we implement a coarse approximation of this procedure, where we think of the number of samples $N$ as a parameter that varies with time and set

$$-N^{-2}\overline{C} \approx \frac{1}{m} \sum_{i=1}^{m} \underset{(x,y)}{\mathbb{E}} \left[ p_{w_t^{(i)}}(y \mid x) - p_{w_{t+\Delta t}^{(i)}}(y \mid x) \right] \tag{17}$$

where $t$ denotes the number of MCMC updates and $\Delta t$ is the interval after which we increase the sample size by one (in practice, this involves simply adding a new sample, or a few samples, to the

data-loader and waiting until the draws from the Markov chain become uncorrelated, e.g., for a few epochs). In practice, we do not sweep all values of $N$ but instead sweep through a set of sample sizes (but still use these finite difference estimates of the thermodynamic quantities). This construction is inspired from the work of Crooks (2007) which estimates the thermodynamic length, a quantity that characterizes the distance between two equilibrium states by endowing the state-space with a Riemannian metric and calculating the length of a path that joins the end points using quasi-static transformations.

The MCMC approach allows us to estimate the average energy $\overline{U}$ and learning capacity $\overline{C}$ using a single Markov chain, but one where the number of samples $N$ increases with time $t$. As Fig. S.1 shows, the estimates of the thermodynamic quantities using this procedure exhibit the same trends that we have discussed in the main paper in Fig. 2. The effective dimension is much smaller than the number of weights in these models. The MCMC-based estimate of the learning capacity is much larger. This can be explained by the large derivative of the estimate of $\overline{U}$ in Fig. S.1 (a,c). It is interesting to note that the MCMC-based estimate of $\overline{U}$ is also larger than the corresponding one using SGD in Fig. 2. We suspect that this is because of the sharper posterior of SGD than stochastic gradient Langevin dynamics; this is consistent with the fact that in practice, SGLD often obtains poorer test errors than SGD. There is another way to interpret the results of this experiment: the MCMC-based learning capacity estimates are higher than those using SGD discussed in the main paper. While we have focused on the utility of the learning capacity for model selection across architectures and datasets, we see some evidence here that the learning capacity can also distinguish between different training algorithms (which affect the kinds of solutions found at the end of training). Again, it is reassuring that *smaller* the learning capacity, better the test loss.

## C   INTERPRETING THE LEARNING CAPACITY USING SINGULAR LEARNING THEORY

One can obtain a more general version of PAC-Bayes bounds above by directly calculating $-\log Z(N)/N$. Watanabe (2009) has calculated asymptotic expressions of this quantity, which is also called stochastic complexity to show that

$$- \mathop{\mathbb{E}}_{D \sim P}[\log Z(N)] = K \log N + \mathcal{O}(\log \log N), \tag{18}$$

where $K$ is the so-called real-log canonical threshold (RLCT) in algebraic geometry that characterizes, roughly speaking, the co-dimension of the model after accounting for symmetries (e.g., permutation symmetries in a neural network) of the weight space. The Fisher Information Metric (FIM) $g(w) \in \mathbb{R}^{p \times p}$ is the metric of the manifold of probability distributions $\{p_w(y \mid x) : w \in \mathbb{R}^p\}$ parameterized by the weights $w$:

$$(g(w))_{kl} = \frac{1}{N} \sum_{i=1}^{N} \sum_{y=1}^{m} p_w(y \mid x_i) \partial_{w_k} \log p_w(y \mid x_i) \partial_{w_l} \log p_w(y \mid x_i). \tag{19}$$

The FIM becomes singular ($\det g(w) = 0$) at these symmetries because there are directions in which the weights can be perturbed without changing the probabilities $p_w(y \mid x)$. Statistical inference of such models, which includes deep networks, Gaussian mixture models, Bayesian networks with hidden units etc., is the subject of singular learning theory. For models without singularities (also called regular models) such as linear regression, the log-canonical threshold is equal to the number of parameters, i.e., $K = p/2$ and, in this form (18), is well-known as the Bayes Information Criterion (BIC). In general, the log-canonical threshold is smaller than half the number of parameters for non-regular models. From (6) and (18), we have that, up to a constant is the entropy of the true labels,

$$- \mathop{\mathbb{E}}_{x,y \sim P}[\log p(y \mid x, D)] = \overline{U}(N) = \frac{K}{N} \tag{20}$$

asymptotically as $N \to \infty$. In other words, smaller the log-canonical threshold $K$, smaller the average test negative log-likelihood. In summary, the concept of model complexity, as captured by the real-log canonical threshold ($K$), plays a crucial role in understanding and evaluating the performance of machine learning models. Models with lower values of $K$, indicating lower complexity, exhibit superior predictive capabilities. From (20), we now have

$$\overline{C}(N) = -N^{-2} \partial_N \overline{U} = K. \tag{21}$$

In other words, the learning capacity is equal to the log-canonical threshold which characterizes the number of degrees of freedom that are constrained in the trained model and thereby controls the negative test log-likelihood in (20). The techniques developed in this paper to calculate the learning capacity can therefore be understood as effective ways to calculate the real-log canonical threshold in algebraic geometry.

**Remark 1.** We can get more intuition on the real-log canonical threshold using an alternate definition. Watanabe (2009, Theorem 7.1) showed that

$$K(w) = \lim_{\epsilon \to 0} \frac{\log V_w(a\epsilon) - \log V_w(\epsilon)}{\log a} \tag{22}$$

for any $a > 0$ where $V_w(\epsilon)$ is the volume under the prior $\varphi$ of the sub-$\epsilon$ level set of the loss, i.e.,

$$V_w(\epsilon) = \int_0^\epsilon \mathrm{d}\epsilon' \int \mathrm{d}w' \, \delta_{\epsilon'}(\hat{H}(w'))\varphi_w(w').$$

The RLCT $K(w)$ can therefore be understood as the rate at which the logarithm of the number of hypotheses (entropy) under consideration increases as we allow an incrementally larger value of the training loss $\hat{H}(w)$ (from $\epsilon$ to $a\epsilon$). One way to understand this is to set $\hat{H}(w) = 1$ and $a = e$, i.e, calculate the number of hypotheses without considering their training loss, and see that the limit in (22) is the logarithm of the volume of the shell of thickness $(e-1)\epsilon$ in $p$ dimensions as the radius of the sphere $\epsilon$ goes to zero; this suggests that the real-log canonical threshold is the number of parameters: $K(w) = p$ up to an additive constant.

**Discussion on symmetries in the weight space** Even if deep networks were to be trained with plentiful data, there would still be anomalies in the number of parameters and the complexity of the learned function purely because of the hierarchical architecture. This is because, for singular models such as deep networks, there are many symmetries in the weight space and multiple weight configurations map to the same predictions (Watanabe, 2007). Geodesics of the manifold of probability distributions accumulate near such singularities and therefore the training dynamics can slow down enormously (Amari et al., 2018; Transtrum et al., 2011). Building upon classical results in statistics (Dacunha-Castelle & Gassiat, 1997), Watanabe has developed a sophisticated theoretical framework based on algebraic geometry to characterize the statistical properties of such singular models. In a different context, it has been noticed that regression models fitted to data from systems biology are "sloppy", i.e., they have few number of tightly constrained stiff parameters and a large number of poorly constrained parameters that do not affect predictions much (Brown et al., 2004; Gutenkunst, 2007). These ideas have also been used to study deep networks and have provided novel insights into the geometry of the hypothesis space of deep networks (Yang et al., 2022b; Mao et al., 2022; Ramesh et al., 2023). Singular learning theory and sloppy models rely on calculating the log-canonical threshold and the Fisher Information Metric to characterize the underlying model—and for modern deep networks trained on large datasets these quantities are rather daunting to compute. The learning capacity discussed here can be used to capture the same concepts and apply them in practice, but it is much easier to compute.

## D    DETAILS OF THE EXPERIMENTAL SETUP

**Datasets** We use MNIST (LeCun et al., 1990) and CIFAR-10 (Krizhevsky, 2009) datasets for our experiments on image classification, use WikiText-2 for experiments on text generation, and use binary-class (miniboone, higgs, numerai) and multi-class datasets (covertype, jannis, connect) for experiments using tabular data; see (Fakoor et al., 2020) for a description of how the tabular datasets were curated. For MNIST, we set up a binary classification problem (even digits vs. odd digits). The synthetic data was created by sampling $d = 200$ dimensional inputs for at most $N = 50,000$ from a Gaussian distribution whose covariance has eigenvalues that decay as $\lambda_i = e^{-\kappa i}$ (i.e., larger the slope $\kappa$, more low-dimensional the inputs) and labeled these inputs using a random teacher network with one hidden layer (1000 neurons, 2 classes). We also use CIFAR10 with noise rate 0.2 to reproduce the double descent phenomenon on ResNet18, see (Nakkiran et al., 2019) for a description of how we add noise to CIFAR10. (Notice: for **??**, the learning capacity is also estimated on the noisy CIFAR10, in order to match the results.)

**Architectures** We use the original LeNet-5 network (convolutional layers and linear layers with tanh as the activation function) and MLPs (one layer, different numbers of hidden neurons and ReLU non-linearity) to train on the MNIST dataset and the synthetic dataset. We use a Wide-ResNet (Zagoruyko & Komodakis, 2016) with a depth of 10 layers and a widening factor of 8 and the ALLCNN network (Springenberg et al., 2015) to train on CIFAR-10. We also train multi-layer RNNs and LSTMs on WikiText-2. For the tabular datasets, we use random forest-based (RFs) and $k$-NN ($k$-nearest neighbor) classifiers. We set the total estimators in RF to be 1000, and use the gini criterion for splitting. We do not restrict the maximal depth of each tree, this entails that each node will expand until the leaves are pure (i.e., all samples in the leaf are from the same class). For the $k$-nearest neighbor classifiers, we set the number of neighbors to be 10.

**Training Procedure**  We use all samples from the training set and record the average energy for 14 different values of $N$ in the set [200, 400, 600, 800, 1000, 2000, 4000, 6000, 8000, 10000, 20000, 30000, 40000, 50000]. The MCMC-based estimates are much more inexpensive computationally and we therefore sampled 63 different values of $N$ uniformly on a log-scale between 100 and 50,000. No data augmentation is used. We only normalize the images of each dataset with mean and standard deviation. As discussed in §2.2, we used $k = 5$ cross-validation folds to estimate the average values of $U, C$. We train all models with SGD and Nesterov's momentum (with coefficient 0.9), and a learning rate schedule that begins with a value of 0.5 and decays using a cosine curve for one cycle (Smith & Topin, 2018). We train for 200 epochs for all datasets and models. The batch size is set to be 64 for all experiments.

Our implementation for WikiText-2 is adapted from pytorch/examples/word_language_model. The only difference is the size of the training dataset. We separate the first $N$ rows of the complete data set as the sub-dataset. The different values of $N$ are [200, 400, 600, 800, 1000, 2000, 4000, 6000, 8000, 10000, 20000, 30000, 40000]. The initial learning rate for multi-layer RNN is 5, and the initial learning rate for multi-layer LSTM is 20. Whenever there is no improvement on the validation dataset, we divide the learning rate by 4.

We will now describe how we add samples incrementally while running SGLD. Suppose we start from some initial sample size $N_1$ and would like to change it to $N_2$. We first split the $N_1$ samples into $k = 10$ equal-sized folds and run $k = 10$ Markov chains in parallel on these folds; these Markov chains are run completely independent of each other. After 20 epochs, we add $(N_2 - N_1)/k$ samples into each of the $k$ Markov chains' dataset; we assume that the Markov chain reaches an equilibrium after 20 epochs of training on this expanded dataset with $((k-1)N_2)/k$ samples and draw 10 samples $w^{(i)}$ from the last 10 epochs to calculate the expressions in (16) and (17).

## E  FURTHER EXPERIMENTAL RESULTS FOR RANDOM FORESTS

We saw in the main text (see Fig. 5a) that the random forest models do not exhibit freezing of the learning capacity at large samples. This is an indicator of the powerful function approximation capabilities of these models. Roughly speaking, this is an attempt to think of the number of leaves in a random forest as being equivalent to the number of weights in a neural network. Both quantities describe the number of degrees of freedom available to their respective models and training corresponds to allocating these degrees of freedom to fit the data. Since we saw that the learning capacity of neural networks exhibits freezing at large sample sizes, we should also expect this behavior for random forests. We set the total number of trees in the forest as 1,000 (same as the original experiment) but restricted the maximal amount of leaf nodes to 100 (in the original experiment, each node is split until the leaves are pure). As shown in Fig. S.3a and Fig. S.3b, the learning capacity of random forest models exhibits freezing at large sample sizes when the number of leaf nodes in each tree is restricted. We therefore believe that freezing is a general phenomenon for machine learning models with a fixed number of degrees of freedom.

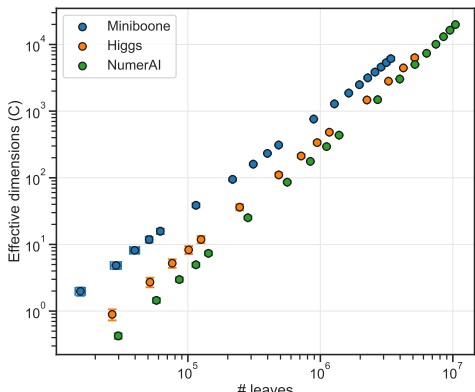

**Figure S.2:** The learning capacity is much smaller (about 0.1%) than the number of leaves in a random forest for all three tabular learning datasets considered here. The capacity also increases monotonically as the number of leaves grows.

## F  A GUIDE TO THE CALCULATIONS IN THE MAIN PAPER

**Basic formulation for the partition function**  We provide detailed calculations for the basic formulation (6)

$$\hat{H}(w) = -\frac{1}{N}\sum_{i=1}^{N}\log p_w(y_i \mid x_i)$$

$$N\hat{H}(w) = -\sum_{i=1}^{N}\log p_w(y_i \mid x_i)$$

(23)

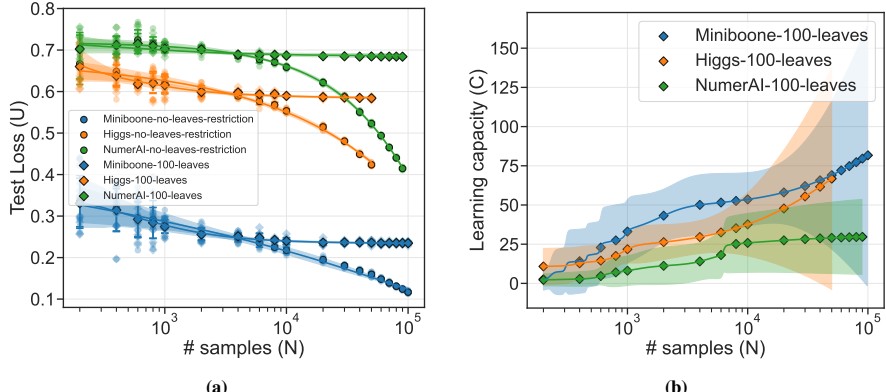

**(a)**                              **(b)**

**Figure S.3: (a)** shows the average energy for random forest models on different datasets (different colors) with (diamonds) and without (circles) a restriction on the number of leaves in each tree. Although the average energy is quite similar for both methods at small sample sizes, as expected models without any restrictions continue to decrease the average energy at large sample sizes. This corresponds to the saturation of the learning capacity in **(b)**; contrast this with Fig. 5a where the learning capacity is growing quickly even at large sample sizes.

$$Z(N) = \int \mathrm{d}w\, \varphi(w) e^{-N\hat{H}(w)}$$
$$= \int \mathrm{d}w\, \varphi(w) e^{\sum_{i=1}^{N} \log p_w(y_i|x_i)} \tag{24}$$
$$Z(N+1) = \int \mathrm{d}w\, \varphi(w) e^{\sum_{i=1}^{N+1} \log p_w(y_i|x_i)}$$

$$\log p(y \mid x, D) = \log \frac{\int \mathrm{d}w\, p_w(y \mid x)\varphi(w) e^{-N\hat{H}(w)}}{\int \mathrm{d}w\, \varphi(w) e^{-N\hat{H}(w)}}$$
$$= \log \int \mathrm{d}w\, p_w(y \mid x)\varphi(w) e^{-N\hat{H}(w)} - \log \int \mathrm{d}w\, \varphi(w) e^{-N\hat{H}(w)}$$
$$= \log \int \mathrm{d}w\, \varphi(w) e^{-N\hat{H}(w)+\log p_w(y|x)} - \log \int \mathrm{d}w\, \varphi(w) e^{-N\hat{H}(w)} \tag{25}$$
$$= \log \int \mathrm{d}w\, \varphi(w) e^{\sum_{i=1}^{N+1} \log p_w(y_i|x_i)} - \log \int \mathrm{d}w\, \varphi(w) e^{\sum_{i=1}^{N} \log p_w(y_i|x_i)}$$
$$= \log Z(N+1) - \log Z(N)$$
$$\approx -U,$$

**Exact calculation of learning capacity for the example of convex energy** $\hat{H}(w) = 1/2 \langle w - w^*, A(w - w^*) \rangle$ **with full rank symmetry Hessian** $A \succ 0$ . For this paragraph, we provide direct calculations for the example provided in the method section. Here we make some assumptions to simplify the calculations. We assume uniform prior ($\varphi(w) = 1$) and $w^* = 0$.

$$Z(N) = \int \mathrm{d}w\, \varphi(w) e^{-N\hat{H}(w)}$$
$$= \int \mathrm{d}w\, e^{-\frac{N}{2} w^T A w} \tag{26}$$

Since A is symmetric, we can write A as $\sum_{i=1}^{p} \lambda_i v_i v_i^T$ where $v_i$ and $\lambda_i$ is the eigenvector and eigenvalue of matrix A. We can also write $w$ as linear combination of eigenvectors of A since A is full rank. ($w = \sum a_i v_i$), and we have the following:

$$Z(N) = \int \mathrm{d}w\, e^{-\frac{N}{2} w^T A w}$$
$$= \int \mathrm{d}a_1 \dots \mathrm{d}a_p\, e^{-\frac{N}{2} \sum_{i=1}^{p} \lambda_i a_i^2} \tag{27}$$
$$= \int \mathrm{d}a_1\, e^{-\frac{N}{2}\lambda_1 a_1^2} \dots \int \mathrm{d}a_p\, e^{-\frac{N}{2}\lambda_p a_p^2} \quad = \prod_{i=1}^{p} \left(\frac{2\pi}{N\lambda_i}\right)^{\frac{1}{2}}$$

$$C(N) = N^2 \partial_N^2 \log Z(N)$$
$$= N^2 \left( \frac{\partial_N^2 Z(N)}{Z(N)} - \left( \frac{\partial_N Z(N)}{Z(N)} \right)^2 \right)$$
$$= N^2 \left( \int dw \, (\frac{1}{2} w^T A w)^2 \frac{e^{-\frac{N}{2} w^T A w}}{z(N)} - \left( \int dw \, \frac{1}{2} w^T A w \frac{e^{-\frac{N}{2} w^T A w}}{z(N)} \right)^2 \right)$$
$$= N^2 \left( \int da_1 ... da_p \, (\frac{1}{2} \sum_{i=1}^p \lambda_i a_i^2)^2 \frac{e^{-\frac{N}{2} \sum_{i=1}^p \lambda_i a_i^2}}{z(N)} - \left( \int da_1 ... da_p \, (\frac{1}{2} \sum_{i=1}^p \lambda_i a_i^2) \frac{e^{-\frac{N}{2}(\sum_{i=1}^p \lambda_i a_i^2)}}{z(N)} \right)^2 \right)$$
$$= N^2 \left( \int da_1 ... da_p \, \frac{1}{4} (\sum_{i=1}^p \lambda_i^2 a_i^4 + \sum_{i,j=1, i \neq j} \lambda_i \lambda_j a_i^2 a_j^2) \frac{e^{-\frac{N}{2} \sum_{i=1}^p \lambda_i a_i^2}}{z(N)} \right.$$
$$\left. - \left( \int da_1 ... da_p \, (\frac{1}{2} \sum_{i=1}^p \lambda_i a_i^2) \frac{e^{-\frac{N}{2}(\sum_{i=1}^p \lambda_i a_i^2)}}{z(N)} \right)^2 \right)$$
$$= N^2 \left( \int da_1 ... da_p \, \frac{1}{4} \sum_{i=1}^p \lambda_i^2 a_i^4 \frac{e^{-\frac{N}{2} \sum_{i=1}^p \lambda_i a_i^2}}{z(N)} \right)$$
$$+ N^2 \left( \int da_1 ... da_p \, \frac{1}{4} \sum_{i,j=1, i \neq j} \lambda_i \lambda_j a_i^2 a_j^2 \frac{e^{-\frac{N}{2} \sum_{i=1}^p \lambda_i a_i^2}}{z(N)} \right)$$
$$- N^2 \left( \int da_1 ... da_p \, \frac{1}{2} \sum_{i=1}^p \lambda_i a_i^2 \frac{e^{-\frac{N}{2}(\sum_{i=1}^p \lambda_i a_i^2)}}{z(N)} \right)^2$$
$$= \frac{3}{4} p + \frac{1}{4} p(p-1) - \frac{1}{4} p^2$$
$$= \frac{p}{2}$$
$$(28)$$

The above provides direct calculation of the learning capacity for regular (non-singular) models.

**Exact calculation of learning capacity for the example of convex energy** $\hat{H}(w) = 1/2 \langle w - w^*, A(w - w^*) \rangle$ **with gaussian prior** .

$$Z(N) = \int dw \, (\frac{\epsilon}{2\pi})^{\frac{p}{2}} e^{-\frac{1}{2} w^T \epsilon I w} e^{-\frac{N}{2} w^T A w}$$
$$= \int dw \, (\frac{\epsilon}{2\pi})^{\frac{p}{2}} e^{-\frac{1}{2} w^T (NA + \epsilon I) w} \tag{29}$$
$$= \prod_{i=1}^p (\frac{\epsilon}{N\lambda_i + \epsilon})^{\frac{1}{2}}$$

$$\log Z(N) = \frac{1}{2} \sum_{i=1}^p \log \left( \frac{\epsilon}{N\lambda_i + \epsilon} \right)$$
$$C(N) = N^2 \partial_N^2 \log Z(N)$$
$$= \frac{1}{2} \sum_{i=1}^p (\frac{N\lambda_i}{N\lambda_i + \epsilon})^2 \tag{30}$$
$$= \frac{p}{2} - \frac{1}{2} \sum_{i=1}^p \frac{2N\epsilon\lambda_i + \epsilon^2}{(N\lambda_i + \epsilon)^2} \quad = \frac{p}{2} - \frac{\epsilon}{N} \sum_{i=1}^p \frac{\lambda_i}{(\lambda_i + \frac{\epsilon}{N})^2} - \frac{1}{2}(\frac{\epsilon}{N})^2 \sum_{i=1}^p \frac{1}{(\lambda_i + \frac{\epsilon}{N})^2}$$

If we have $\lambda >> \frac{\epsilon}{N}$ and $p \approx N$, we can approximate the results.

$$C(N) = \frac{p}{2} - \epsilon \text{HM}^{-1}(\lambda_i(A)) \tag{31}$$

**Exact calculation** The following provides exact calculation of the generalization error upper bound.

$$-\frac{\log Z(N)}{N} \approx \frac{p \log N}{2N} + \frac{\sum_i \log \lambda_i}{2N} + \sum_i \frac{\epsilon}{2N\lambda_i} + \frac{\epsilon \|w^* - w_0\|^2}{N}$$

$$= \frac{p + \epsilon\, \mathrm{HM}^{-1}(\lambda_i)}{2N} + \frac{\sum_i \log \lambda_i}{2N} + \frac{\epsilon \|w^* - w_0\|^2}{N} \tag{32}$$

$$= \frac{C}{N} + \frac{\epsilon \|w^* - w_0\|^2}{N}, \quad \text{from (5)},$$

$$Z = \int \mathrm{d}w \left(\frac{\epsilon}{2\pi}\right)^{\frac{p}{2}} e^{-\frac{\epsilon}{2}(w-w_0)^T I (w-w_0)} e^{-\frac{N}{2}(w-w^*) H(w^*)(w-w^*)}$$

Let $Q = w - w^*$,

$$= \int \mathrm{d}Q \left(\frac{\epsilon}{2\pi}\right)^{\frac{p}{2}} e^{-\frac{\epsilon}{2}(Q+w^*-w_0)^T I (Q+w^*-w_0)} e^{-\frac{N}{2} Q H(w^*) Q}$$

$$= e^{-\frac{\epsilon}{2}(w^*-w_0)^2} \left(\frac{\epsilon}{2\pi}\right)^{\frac{p}{2}} \int \mathrm{d}Q\, e^{-\frac{\epsilon}{2}(Q+w^*-w_0)^T I (Q+w^*-w_0)} e^{-\frac{N}{2} Q H(w^*) Q} \tag{33}$$

$$= e^{-\frac{\epsilon}{2}(w^*-w_0)^2} \left(\frac{\epsilon}{2\pi}\right)^{\frac{p}{2}} \int \mathrm{d}Q\, e^{-\frac{1}{2}(Q^T (NH(w)+\epsilon I)Q + 2\epsilon(w^*-w_0)^T Q)}$$

Projection to eigenvector coordinate and let $b_i$ be the components of $w^* - w_0$

$$= e^{-\frac{\epsilon}{2}(w^*-w_0)^2} \left(\frac{\epsilon}{2\pi}\right)^{\frac{p}{2}} \int \mathrm{d}a_1 ... \mathrm{d}a_p\, e^{-\frac{1}{2}\sum_{i=1}^{p}(N\lambda_i+\epsilon)a_i^2 + 2b_i a_i} \tag{34}$$

let $c_i = \frac{\epsilon b_i}{N\lambda_i + \epsilon}$

$$= e^{-\frac{\epsilon}{2}(w^*-w_0)^2} \left(\frac{\epsilon}{2\pi}\right)^{\frac{p}{2}} e^{\frac{\epsilon^2}{2}\sum_{i=1}^{p} \frac{b_i^2}{N\lambda_i+\epsilon}} \int \mathrm{d}a_1 ... \mathrm{d}a_p\, e^{-\frac{1}{2}\sum_{i=1}^{p}(N\lambda_i+\epsilon)(a_i^2+c_i)}$$

$$= e^{-\frac{\epsilon}{2}(w^*-w_0)^2} e^{\frac{\epsilon^2}{2}\sum_{i=1}^{p} \frac{b_i^2}{N\lambda_i+\epsilon}} \prod_{i=1}^{p} \left(\frac{\epsilon}{N\lambda_i + \epsilon}\right)^{\frac{1}{2}} \tag{35}$$

And we have the following

$$-\frac{\log Z(N)}{N} = \frac{\epsilon}{2N}(w^* - w_0)^2 + \frac{1}{2N}\sum_{i=1}^{p} \log \frac{N\lambda_i + \epsilon}{\epsilon} + \frac{\epsilon^2}{2N}\sum_{i=1}^{p} \frac{b_i^2}{N\lambda_i + \epsilon} \tag{36}$$

Consider the case where $\lambda >> \epsilon$ and we can have

$$= \frac{\epsilon}{2N}(w^* - w_0)^2 + \frac{1}{2N}\sum_{i=1}^{p} \log \frac{N\lambda_i}{\epsilon} - \frac{\epsilon^2}{2N^2}\sum_{i=1}^{p} \frac{b_i^2}{\lambda_i}$$

$$= \frac{\epsilon}{2N}(w^* - w_0)^2 + \frac{p}{2N} \log \frac{N}{\epsilon} + \frac{1}{2N}\sum_{i=1}^{p} \log \lambda_i - \frac{\epsilon^2}{2N^2}\sum_{i=1}^{p} \frac{b_i^2}{\lambda_i} \tag{37}$$

If we choose $\epsilon$ small enough, we can have

$$\frac{C}{N} + \frac{\epsilon \|w^* - w_0\|^2}{2N} \approx -\frac{\log Z(N)}{N} \tag{38}$$

Upper bound of the generalization is the upper bound of the complexity term. Minimizing the objective is equivalent to minimizing the complexity of the Bayesian models.

## G Q & A

**Q: What are the practical implications of this work?** The primary focus of this paper is theoretical but it indeed also has some very broad practical implications. Perhaps the most important one (in our mind) is as follows. The learning capacity can be estimated using a cross-validation-like procedure to calculate the derivative of the test loss with respect to the number of samples. One can estimate the saturation of the learning capacity using its second derivative to obtain crisp guidelines as to whether one should invest in procuring more data or whether the test loss would improve if the model architecture/class were different. This is important because it is increasingly being noticed that large models are significantly under-trained Hoffmann et al. (2022). The learning capacity is a very good

way to check how much of the capacity of the model is being utilized. In this sense, the utility of learning capacity is akin to that of scaling laws in language models. Scaling laws tell us how much more data to procure or how long to train for, and we can use the learning capacity to ask similar questions. One could also use learning capacity as an architecture search procedure; this is similar to Yang et al. (2022a) who used small networks to search for the training hyper-parameters of bigger networks.

By eliminating the double descent phenomenon, we believe we have brought deep networks back into the classical realm of analysis in machine learning. We can think of them as another model in the bag of models that a practitioner has. In fact, we have also brought other models such as random forests and $k$-nearest neighbor classifiers on the same playing field. The learning capacity allows us to think about all these models using the same set of ideas, and in a theoretically rigorous fashion. This has broad implications for model selection, both in practice and in theory.

**Q: What are the theoretical implications of this work?**    The fact that deep networks predict extremely well on new data in spite of often having many more parameters than the number of training data is one of the biggest open questions today. There is a large body of work that tackles this question but a definitive explanation is yet to be found. Our paper adds a different perspective to the literature.

We use a very different set of tools, from thermodynamics and statistical physics, to motivate a quantity called learning capacity (which is analogous to the heat capacity of a material) that can be thought of as the effective number of degrees of freedom that are constrained by the data in a trained deep network. The smaller the number of these constrained degrees of freedom, the fewer the number of parameters that the network is actually using to make its predictions and therefore even if the architecture itself has a large number of other parameters, we should calculate the effective dimensionality using only the parameters that play a role in making the predictions. We elaborated upon rigorous connections of this idea to existing results in learning theory (e.g., PAC-Bayes theory, singular learning theory). There are also some other ones that we can glean, e.g., sharpness-based measures of generalization (Foret et al., 2020) can be seen to be related to learning capacity using the calculation in (5), or techniques to find and study wide regions in the energy landscape (Chaudhari et al., 2017; Baldassi et al., 2016) are already closely related to the log-partition function (they calculated $\log Z$ using MCMC locally in the neighborhood of a solution) and PAC-Bayes bounds (Dziugaite & Roy, 2017).

We found strong evidence that the learning capacity captures generalization in deep learning correctly (e.g., it is much smaller than the number of weights, correlates well with the test loss, matches numerically tight PAC-Bayes generalization bounds). There are also surprising findings, e.g., if we plot the test loss against the learning capacity, then the double descent phenomenon vanishes. This suggests that it may be fruitfully used to understand the complexity of functions learned by deep networks.

**Q: Can you calculate learning capacity other models, datasets and input modalities?**    This is a very interesting point. While existing concepts for studying generalization are rather abstract and require complicated implementations (and fall short of unraveling the mystery), the learning capacity is a simple quantity that is also easy to estimate. Any model (even a Python function!) can be used to calculate the learning capacity.

In terms of correlation of the learning capacity with the test loss, our results are comparable (often better) than those from exhaustive comparisons such as Jiang et al. (2019) and directly comparable to rigorous ones such as Yang et al. (2022b). This is strong evidence that the learning capacity is useful to understand generalization.

**Q: How is the learning capacity in this paper different from the original work of LaMont & Wiggins (2019)?**    Our paper is strongly inspired by theirs. Lamont and Wiggins coined the term "learning capacity" and showed that for some systems (a free particle in vacuum and Gaussian distributions), the learning capacity is proportional to the number of degrees of freedom. They suggested that one could calculate $\overline{U}$ using finite differences of the log-partition function (note, estimating its derivative which is the learning capacity $\overline{C}$ robustly is the difficult step in this approach).

We used the learning capacity to investigate the generalization of machine learning models. All our findings from Section 2.2, especially all the insights discussed in the "Results" section are new. To glean these insights, we made the analogy of Lamont and Wiggins more rigorous, by drawing out a clean connection PAC-Bayes theory (this is critical because it shows that what is essentially a pattern matching exercise is meaningful) and to singular learning theory (in their paper, it was more of a speculation). We have developed an effective procedure to numerically calculate the learning capacity (which is absent in the original paper).

We should emphasize that heat capacity is a very basic thermodynamic quantity. Learning capacity is completely analogous, except that we interpret the statistical ensemble in physics as all possible models. The ideas from thermodynamics used in this paper are in standard undergraduate textbooks on the subject. In particular there is nothing new that we say up to Section 2.1. The correspondence between the number of samples and the inverse temperature is simply because the factor of $1/N$ in (1) shows up in the exponent in (2).

**Q: How can you justify the definition of the learning capacity?** The definition of the learning capacity is simply $C = -N^2 \partial_N \log Z(N)$. As such, it is well-defined as soon as we set up the statistical inference problem. The utility of a definition, and the merit of developing it, lies in how well the definition can be used to explain observations (e.g., test loss), glean new conclusions (e.g., freezing phenomena) and whether it can be interpreted in the light of existing ideas in the field (e.g., PAC-Bayes theory and singular learning theory).

**Q: This paper is difficult to understand** This paper touches a lot of non-standard topics; there are also a lot of technicalities involved (e.g., we are using data-distribution dependent PAC-Bayes bounds and computing a notion of effective dimensionality from those bounds); we have also resorted to approximations (e.g., calculating the log-partition function using a quadratic approximation of the training loss interpret the learning capacity). We have attempted to make the narrative as clear as possible, but it is difficult to explain some of these ideas from scratch in limited space (e.g., Appendix C summarizes an entire chapter from the book on algebraic geometry by Watanabe (2009)).

**Q: Does the learning capacity depend upon the training method? Does changing the training method yield different results?** The learning capacity defined on (4) does not depend upon the method used to train the model (it is an average over the entire weight space). This is just like the VC dimension, which does not depend upon how we train the model.

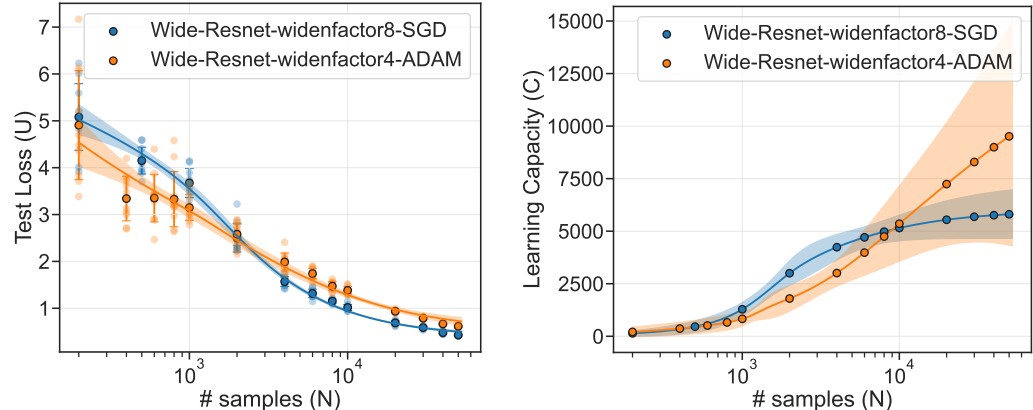

**Figure S.4:** Average energy and learning capacity for two wide residual networks with different widening factors training with SGD and Adam respectively on CIFAR-10. It is remarkable that the learning capacity in the two is quite similar across different sample sizes, and as is the test loss.

The procedure to numerically estimate the learning capacity does depend upon the training method. And different experiments in the paper use different training methods, e.g., SGD for training each deep network in Monte Carlo, fitting the random forest using bootstrapping, or the MCMC method for estimating learning capacity in Appendix B. In all cases, we have shown that the conclusions drawn using the learning capacity are both consistent with existing results in the literature on generalization (e.g., correlation to test loss) and also lead to some surprising phenomena which have not been noticed yet. We additionally show one more experiment in Fig. S.4 where we trained two different wide residual networks on CIFAR-10 using two training methods (SGD vs. Adam); the learning capacity in both cases is very similar across a wide range of sample sizes $N$, and as is the test loss. This suggests that our estimate of learning capacity is insensitive to the details of the training procedure.

We have also used a broad diversity of architectures, datasets and input modalities and obtained conclusions that are consistent across the spectrum. This suggests that we can not only instantiate the learning capacity in different settings but also implement it robustly.

