# OpenReview forum: "Investigating the effective dimensionality of a model using a thermodynamic learning capacity"
_ICLR.cc/2024/Conference — Submitted to ICLR 2024_

### Official Review · Reviewer_nwdU · 2023-10-18

**Soundness:** 1 poor
**Presentation:** 2 fair
**Contribution:** 3 good
**Rating:** 3
**Confidence:** 3

**Summary:**

This paper provides a candidate as a novel measure of the effective parameters of a machine learning model. It leverages ideas from the thermodynamics literature to argue that the idea of heat capacity can be used as a measure of learning capacity. An explicit formulation of this idea is derived from which an algorithm is derived that is applicable to any learned function. This measure is linked to PAC-Bayesian theory. Several experiments are performed which apply this approach and attempt to investigate well-known challenges (e.g. double descent, model selection) through this lens.

**Strengths:**

* This work addresses the pertinent challenge of measuring the _effective_ parameters used by a model which has important implications across machine learning.
* It considers well-established ideas from the thermodynamics literature which offers a fresh perspective. I found this perspective interesting and generally appreciate it when ideas from other fields are applied in the machine learning context.
* The authors correctly associate a good measure of effective parameters with several important utilizations including model selection and generalization bounds.
* I appreciated the authors efforts to provide additional background and context on some of the concepts introduced in the appendix e.g. App. A.
* The authors attempt to link ideas from thermodynamics with PAC-Bayesian theory. This connection seems interesting to pursue.

**Weaknesses:**

Overall, my main concern with this paper is that I didn't think it sufficiently convinced that the proposed measure of effective parameters was a good one. This was further complicated by the paper attempting to fit too many ideas/claims that were not individually investigated to their conclusion with sufficient rigor. Given that this paper attempts to provide a general-purpose measure of effective parameters with little to no limitations (a very significant claim) I cannot recommend acceptance in its current form based on the arguments and the evidence provided at this stage. However, I hope that the authors are not discouraged as there does seem to be some interesting ideas in this paper that may well result in a valuable contribution after further refinement and critical analysis. I provide my detailed critiques below.

Section 2 - Methods
* While I understand the authors were inspired by notions from thermodynamics and are using results from that field I don't understand the benefit of presenting the method in these terms. I found the reasoning of this section more challenging to follow than necessary due to the switching back and forward. It would be much clearer to simply discuss the quantities of interest in the ML setting (e.g. negative log-likelihood) and derive an expression of effective parameters from there (using results from thermodynamics when required but in the same notation to avoid switching back and forward).
* Why is setting the Boltzmann constant to 1 a reasonable choice? This does not seem to be its true known value and therefore $N$ is not exactly $T^{-1}$. This has downstream effects by my understanding since the authors explicitly interpret the effective parameters relative to the number of parameters stating "the learning capacity is a tiny fraction (∼ 2%) of the total number of weights".
* The four bullet points attempt to argue that learning capacity is a good measure of effective parameters (I would suggest convincing the reader on this point is fundamental to the remainder of the paper). However, this is not a convincing argument in my opinion and I struggled to even understand the reasoning in some cases. For example "[...] the learning capacity C equals half the effective number of parameters K/2." How do we know the true effective parameters here, this is not defined clearly? Why is H defined as it is here (this convex expression) and why should this be convincing?
* In eqn (6) it's not clear to me that the two log terms are a good approximation for -U. The justification is just a statement that it is. Especially since in eqn (7) this approximation is assumed to be good for every data point in the distribution. Why the subtle change to N-1 in eqn (8)?
* The rightmost equality is eqn (8) is unclear to me. Why is this an equality, I understood the LOOCV estimator was approximating something here. Could this step be spelled out further please?
* Then in practice LOOCV is not actually used, k-fold is used instead. There is no analysis of the effect of increasing the value of k. This is another approximation being introduced that requires further consideration.
* The process described at the start of Sec 2.2 (sampling datasets, cross-validation, etc.) should be described as a formal algorithm somewhere.
* The authors report taking 4 bootstrap samples of the data. What exactly is this intended to approximate? Is this supposed to account for the $\mathbb{E}_{x,y \sim P}$?
* At this point it's not entirely clear to me what calculating $\bar{C} = - N^2 \nabla_N \bar{U}$ means exactly. Are you trying to take the gradient of eqn (9) wrt the number of samples N?
* "we used a polynomial mode" - Initially thought the authors meant that the underlying model was polynomial (which it technically could be since its a single hidden layer MLP). They should specify that they are referring to a model being fit to U and C values.
* "this procedure also gives an estimate of the uncertainty" - These estimates are unlikely to be valid I would imagine since the observations are not independent.
* It's claimed that the method "works reasonably well (Fig 1)". How can we tell this is working well from these figures? We don't have a ground truth so presumably this means it produces a shape that the authors hoped? This seems like a weak evaluation.
* "We therefore fit a sigmoid [...]" - Does this not undermine the fourth contribution of the paper that "The learning capacity saturates for very small (equivalent to high temperature) and very large (equivalent to low temperature) sample sizes N ." Since you are effectively enforcing this by modeling it as a sigmoid?

Section 3 - Connection to PAC-Bayesian framework
* While the idea of making this connection to the PAC-Bayesian framework is interesting, I found the execution could have been made more clear. I think this section could have benefitted from a more formal presentation of the equivalence (e.g. a theorem). As with the previous section, including all theory as part of a long discussion section with extensive notation and several steps being insufficiently explicit harms the paper as a whole. I found it difficult to judge just how strong the connection claimed in this section really is.

Section 4 - Experiments
* Even setting aside my aforementioned concerns with a proposed measure of effective dimensionality, I am skeptical that the actual raw value rather than the relative values it outputs should be considered accurate. Comparing this value to the actual number of parameters seems problematic given the number of approximations taken in its estimation.
* The authors mention that this low effective parameter count is consistent with other aspects of the literature e.g. distillation. This would seem to be a testable hypothesis. If we trained a large model with a given effective parameter count and nearly perfectly distill it into a much smaller model, the number of effective parameters should remain constant or decrease very slightly.
* Without being convinced by the underlying measures U and C, the results in Table 1 are insufficiently granular to convince the reader of the effectiveness of these measures. I also find it surprising that regularization wasn't a primary consideration in attempting to validate the metrics.
* It seems that the general intention of these experiments is to investigate the effective parameters of models, datasets, etc. assuming we have a valid measure of effective parameters. I think the overarching problem is that the proposed measure may not be a good choice.
* I found the double descent results in Sec 3.2 suggested issues with the proposed measure. Firstly, I don't see a double descent shape in the LHS plot. Then the right-hand plot seems (a) extremely noisy and (b) does not capture the correct U-shape that we would expect if it was working as we would hope. This seems to me to provide evidence that the measure is _not_ working effectively.
* In Sec 3.3 and Fig 4, in contrast to the authors, I would find it hard to conclude that there is a particularly strong connection based on the results provided in the plot. I also find it hard to believe that the Pearson Correlation between the solid and dashed green lines is 0.99 based on visual inspection.
* No code was provided to inspect the experiments performed in more detail.

Section 5 - Related work
* The related work section seemed to miss all of the existing work of effective parameters, largely from the statistical learning literature from where the concept emerged to the best of my knowledge. See e.g. [1] for a recent attempt in the deep learning context or [2] for a general overview of the concept across various traditional learning models.

Minor
* Using asterisk symbols for footnotes should be avoided since this symbol is also used in the mathematical notation. I would suggest sticking to the standard approach of numbering footnotes.
* The style guide paragraph spacing seems to have been removed to fit more content into the paper. This negatively affects the readability of the work.

Finally, I would briefly note that some of the issues I have raised were also included by the authors in the Q&A section of App F. Specifically (a) "How can you justify the definition of the learning capacity?" and (b) "This paper is difficult to understand". I didn't find the answers provided have resolved these issues. For (a), I don't think this is a self-evidently good measure as the authors imply. For (b), while I appreciate that expressing complex ideas and connections can be challenging, it must be done to a sufficient level to convince the reader of the claims being made even so.


[1] Maddox, W. J., Benton, G., & Wilson, A. G. (2020). Rethinking parameter counting in deep models: Effective dimensionality revisited. arXiv preprint arXiv:2003.02139.

[2] Hastie, T., Tibshirani, R., Friedman, J. H., & Friedman, J. H. (2009). The elements of statistical learning: data mining, inference, and prediction (Vol. 2, pp. 1-758). New York: springer.

**Questions:**

See weaknesses.

---

> ### Author Response · Authors · 2023-11-23
> **Reponse to Reviewer nwdU**
>
> Thank you for your time and detailed review of our paper. We appreciate your attention to detail and the desire to improve the quality of our paper. A lot of the questions that you had asked were minor technical details which we have clarified in the responses below. We have elaborately addressed your major concerns regarding the learning capacity. If these responses are satisfactory, we would encourage you to increase your original score, which we felt was not commensurate with your concerns.
>
> This is not at all a speculative paper, it has a lot of theoretical and empirical depth; and it does make meaningful process upon the problem. Papers such as this are bring in fresh ideas to work on long-standing problems in deep learning and we need all these diverse perspectives if we are to make progress upon the problem. When someone defines a new quantity, there is of course going to be a lot of criticism and questioning, we completely appreciate that. The merit of a quantity like learning capacity is that it is so general and broadly applicable...from parametric models such as deep networks (MLPs, CNNs, LSTMs etc.) to even non-parametric models such RFs and k-NNs; it is well defined even for a Python function that makes predictions! As other researchers in the field study these ideas further, we will develop a deeper understanding of such definitions. This paper provides sufficient evidence that the learning capacity is a well-defined and meaningful quantity that can help us understand generalization and can also be estimated accurately.
>
> **While I understand the authors were inspired by notions from thermodynamics and are using results from that field I don't understand the benefit of presenting the method in these terms. I found the reasoning of this section more challenging to follow than necessary due to the switching back and forward. It would be much clearer to simply discuss the quantities of interest in the ML setting (e.g. negative log-likelihood) and derive an expression of effective parameters from there (using results from thermodynamics when required but in the same notation to avoid switching back and forward).**
>
> This is a reasonable comment and we will consider modifying the narrative to make it more friendly to the machine learning audience. The thermodynamics-specific comments in the paper are actually quite few and one can simply think of these quantities as definitions without worrying about their meanings in thermodynamics.
>
> The original purpose of this writing is trying to provide gradual understanding of the context. For the first paragraph, we give simple example to help people understand or have a grasp of the importance of learning capacity. For the second paragraph, we give a much formal formulation and framework about how calculations are done through the process and how it is related to different generalization calculations. There are also some compromises between going to the detail without resorting to appendix since the context have connection to different areas. Thank you for your advice about arrangement of the context and we will change it and explore different ways to make it clear.
>
> **Why is setting the Boltzmann constant to 1 a reasonable choice? This does not seem to be its true known value and therefore is not exactly. This has downstream effects by my understanding since the authors explicitly interpret the effective parameters relative to the number of parameters stating "the learning capacity is a tiny fraction (∼ 2%) of the total number of weights".**
>
> Boltzmann constant is often set to 1 in theoretical physics, there it simply means that we are measuring temperature in the units of energy. It is important for our purposes here, we can simply think of the partition function in Equation 2 as the point of beginning and derive everything from it. Please see Appendix F where we provide more details now. Just to clarify for the Reviewer, the fact that learning capacity is small (which is what we are sating when we say that the learning capacity is a tiny fraction of the number of weights) is not something that depends upon the units in which temperature is measured. Things in physics are quite similar, heat capacity of water is large compared to steel---a statement that can be made independently of the units of temperature.

---

> ### Author Response · Authors · 2023-11-23
> **Responses for nwdU**
>
> **The four bullet points on Page 2 argue that learning capacity is a good measure of effective parameters (I would suggest convincing the reader on this point is fundamental to the remainder of the paper). However, this is not a convincing argument in my opinion and I struggled to even understand the reasoning in some cases. For example "[...] the learning capacity C equals half the effective number of parameters K/2." How do we know the true effective parameters here, this is not defined clearly? Why is H defined as it is here (this convex expression) and why should this be convincing?**
>
> We believe there is a gap in the understanding here. There are many quantities to estimate the complexity of a model, e.g., VC-dimension, Rademacher complexity, PAC-Bayes bounds etc. These quantities are all characterizing the effective dimensionality of a model in different ways, e.g., generalization gap <= sqrt (VC-dim/n). In our case, we adopt the definition of “true effective parameters” proposed by Watanabe (refer to Algebraic Geometry and Statistical Learning Theory). If a linear model has p parameters then the VC-dimension is p; therefore the effective dimensionality of such a model is p. For some models, e.g., singular models discussed in Appendix C, the effective dimensionality may be smaller. In information theory, there are more quantities such as minimum description length, Bayesian information criterion etc. See the first paragraph of the Discussion for references.
>
> The biggest issue in deep learning is that these quantities are all very large, they scale with the number of weights in the network. So the existing theories of generalization are rendered vacuous. Effectively, in deep learning these quantities, even if they do tell us some general principles, do not seem to correlate well with the test loss. The learning capacity does correlate with the test loss, so it mitigates this issue. The second issue is that some of the standard learning theoretic quantities are difficult to estimate for deep networks, e.g., PAC-Bayes bounds work well but it is very expensive to calculate them. The learning capacity is an approximation of PAC-Bayes-based effective dimensionality (Equation 13) and easier to calculate.
>
> H is not defined in the four bullets, it is defined in Equation 1. In the bullets, we have simply given examples of how the learning capacity looks like for different values of H and the prior. This helps the reader appreciate why we call C the effective dimensionality of the model.
>
>
>
>
>
> **In eqn (6) it's not clear to me that the two log terms are a good approximation for -U. The justification is just a statement that it is. Especially since in eqn (7) this approximation is assumed to be good for every data point in the distribution. Why the subtle change to N-1 in eqn (8)?**
> ** **
>
> The definition of U is given in Equation (4). The derivation in Equation (6) is a finite-difference approximation of U. Equation 7 follows from 6 directly due to linearity of expectation. There is no reason for changing to N-1 in Equation (8), it just makes it easy to write the expression of Ubar as the standard leave-one-out estimator (LOOCV) for a dataset of N samples. See Appendix F for more details.
>
> **The rightmost equality is eqn (8) is unclear to me. Why is this an equality, I understood the LOOCV estimator was approximating something here. Could this step be spelled out further please?**
>
> The right most equality of Equation (8) follows directly from the definition of the partition function. We now provide more details in Appendix F.
>
> **The process described at the start of Sec 2.2 (sampling datasets, cross-validation, etc.) should be described as a formal algorithm somewhere.**
>
> The narrative in Section 2.2 is the algorithm right? It describes each step of the calculation in complete details.

---

> ### Author Response · Authors · 2023-11-23
> **Responses for nwdU**
>
> **The authors report taking 4 bootstrap samples of the data. What exactly is this intended to approximate? Is this supposed to account for the E_{x,y ~P}?**
>
> No E_{x,y ~P} is estimated directly when we calculate the average test loss over the validation fold. The bootstraps are done for a different reason. The quantity Z(N) is a random variable because it depends upon the samples in the training dataset; for different draws of the training dataset with the same N, we will have different values of Z(N). In order to improve our estimate of the learning capacity, i.e., to ensure that the estimate is not a function of the specific samples in the training dataset but only of the number of samples N, we do a bootstrap. This is a very common application when we are estimating quantities that depend on the realization of the particular training set.
>
> ** Then in practice LOOCV is not actually used, k-fold is used instead. There is no analysis of the effect of increasing the value of k. This is another approximation being introduced that requires further consideration.**
>
> We can see the k-fold experiment as an approximation of LOOCV. We train on the k-1 folds of data, and calculate the average test loss on the remaining 1 fold to approximate the result of LOOCV.  It is very expensive to estimate LOOCV in practice. As k gets closer to N, the results from k-fold will be get closer to LOOCV. If k equals N, the N fold is exactly LOOCV. The analysis of how the Monte Carlo error behaves as a function of k is quite standard, see for example https://theory.stanford.edu/~sergei/papers/ics11-cv.pdf. The variance of the estimator using k-fold goes down as ~1/k compared to LOOCV.
>
> **At this point it's not entirely clear to me what calculating C = -N^2 grad Ubar means exactly.**
>
> To calculate C, we are taking the gradient of Ubar with respect to the number of samples N. This gradient is calculated using finite differences as shown in Equation 8.
>
> **Are you trying to take the gradient of eqn (9) wrt the number of samples N?**
>
> No no! The calculation in Eqn(9) is calculating Ubar by averaging over the test loss over k-folds. The test loss is averaged over all samples inside the remaining folds which is where the n comes from. We either fit it with polynomials or the integration of the sigmoid function and take derivatives to calculate the learning capacity.
>
> **"We used a polynomial model" - Initially thought the authors meant that the underlying model was polynomial (which it technically could be since its a single hidden layer MLP). They should specify that they are referring to a model being fit to U and C values.**
>
> Easy to fix, we will make it clear.
>
> **"this procedure also gives an estimate of the uncertainty" - These estimates are unlikely to be valid I would imagine since the observations are not independent.**
>
> This is not true. The observations that are used to calculate the estimate of uncertainty are obtained from bootstrap, random weight initializations and multiple folds. They are correct numerical estimates of the uncertainty when the assumptions of bootstrap are valid, which are rather broad.
>
> **It's claimed that the method "works reasonably well (Fig 1)". How can we tell this is working well from these figures? We don't have a ground truth so presumably this means it produces a shape that the authors hoped? This seems like a weak evaluation.**
>
>
> “We therefore fit a sigmoid [...]" - Does this not undermine the fourth contribution of the paper that "The learning capacity saturates for very small (equivalent to high temperature) and very large (equivalent to low temperature) sample sizes N ." Since you are effectively enforcing this by modeling it as a sigmoid?
> Responses: The model is fit on U values with explicit form described in eqn(10). We experimentally found that the taking derivatives of polynomial fitting of the U curves show large and small sample size saturation in general. The drawback of polynomial fitting is that we need to specify the degree of polynomials which can be subjective, and the damping of polynomials fitting is inevitable. However, based on the observation we have, if we fit the U curve with the eqn(10), we can prevent the overfitting or underfitting due to subjective choice of degree of the polynomial and we also capture the characteristic of most of the characteristics showed in the experiment data if we assume smooth change in the U and C curves.

---

> ### Author Response · Authors · 2023-11-23
> **Responses for nwdU**
>
> **While the idea of making this connection to the PAC-Bayesian framework is interesting, I found the execution could have been made more clear. I think this section could have benefitted from a more formal presentation of the equivalence (e.g. a theorem). As with the previous section, including all theory as part of a long discussion section with extensive notation and several steps being insufficiently explicit harms the paper as a whole. I found it difficult to judge just how strong the connection claimed in this section really is.**
>
> We are a bit puzzled at the point that the Reviewer is making here. The equivalence between the learning capacity cannot be a theorem! We are simply saying that the expression of the learning capacity can be simplified to be the expression of the effective dimensionality given by PAC-Bayes calculation. This is indeed a “strong connection” that the Reviewer is looking for. Because it is not derived from ad hoc assumptions or imprecise derivations, it is a direct calculation. This connection says that even if the expression for the learning capacity was motivated from thermodynamics and, a priori, doesn't have anything to do with learning theory, this expression can be connected directly established statistical learning theory. It is therefore a meaningful connection.
>
> **Even setting aside my aforementioned concerns with a proposed measure of effective dimensionality, I am skeptical that the actual raw value rather than the relative values it outputs should be considered accurate. Comparing this value to the actual number of parameters seems problematic given the number of approximations taken in its estimation.**
>
> This is not true. The learning capacity is a number that we have proposed as an estimate of the effective dimensionality. This number correlates with the test loss, is much smaller than the number of weights (as we would expect any estimate of the capacity of deep networks to be) and has many other interesting properties. The readers are free to use this number, use relative values if they want to model selection. We do not understand how the Reviewer can conclude that the actual values of the effective dimensionality are not accurate. We defined it, they are accurate by definition! The merit of a definition is whether it is useful, which we have shown that it is.
>
> **The authors mention that this low effective parameter count is consistent with other aspects of the literature e.g. distillation. This would seem to be a testable hypothesis. If we trained a large model with a given effective parameter count and nearly perfectly distilled it into a much smaller model, the number of effective parameters should remain constant or decrease very slightly. Without being convinced by the underlying measures U and C, the results in Table 1 are insufficiently granular to convince the reader of the effectiveness of these measures. I also find it surprising that regularization wasn't a primary consideration in attempting to validate the metrics.**
>
> This is an interesting idea. We believe it will also work. A partial evidence for whether it will work is found in Fig. 3. Each of the colored lines are networks with many different parameters trained on the same number of samples. For the brown line in both Fig 3a and 3b, which are networks trained on all the training data in MNSIT and CIFAR-10, the learning capacity is essentially constant as a function of the number of weights in the model.
>
> Regularization is a tricky thing to do while working with even standard PAC-Bayes bounds. The effect of regularization interacts in complicated ways with the PAC-Bayes optimization. The paper by Jiang et al. (2019) also found that regularization does not affect the test error much in many cases. For example, one natural choice of regularization is the L2 norm. The L2 norm can only restrict the parameters from being too large. It did not prevent the learning capability from being larger or smaller so it is hard to have a dicisive conclusion from this experiment. L2 regularization is not actually that effective for deep learning, see https://arxiv.org/abs/1905.13277. More formally, L2 norm is meaningful for models where the classical bias-variance tradeoff holds. For neural networks with symmetries, the origin is not any more special than some other point in the weight space. For example, there is a scale symmetry in networks with batch-normalization and therefore any two weight configurations along a ray that intersects the origin express the exact same function.
>
> We did perform some experiments using different learning rates, see the last question in the FAQ in Appendix G.
>
> Now that we have defined this quantity and provided an easy way to estimate for deep networks, other researchers and evaluate whether it is useful to understand regularization.

---

> ### Author Response · Authors · 2023-11-23
> **Responses for nwdU**
>
> **It seems that the general intention of these experiments is to investigate the effective parameters of models, datasets, etc. assuming we have a valid measure of effective parameters. I think the overarching problem is that the proposed measure may not be a good choice.**
>
> This is an appropriate characterization of the paper. We have defined the learning capacity and studied it under various settings to see if it works well, e.g., in terms of its correlation to the test loss, its utility to see/unsee double descent, for different non-parametric models etc. We are not assuming in these investigations that our estimate of the effective dimensionality is correct. That part comes from the theory, e.g., the PAC-Bayes calculations, and experiment where we directly checked the learning capacity against PAC-Bayes effective dimensionality (Fig 4).
>
> So far, in literature, the definitions of effective dimension are based on the analysis of the generalization performance of the models. For example, VC dimension and effective dimension mentioned in Yang et al.’s work. They first provide generalization upper bound and collect part of the terms and define them as effective dimension. However, the upper bound can be loose or vacuous due to many assumptions or approximations and this can lead to an overestimation of the effective dimension. In our work, the goal is a direct estimation of the effective dimension without layers of assumptions. The definition is sound and we also provide calculation and experimental results to link to PAC-Bays theorem, which shows similar result with less assumption or approximation and is more general. If you find the effective dimension under PAC-Bays framework meaningful, then our work is just a cheaper way to calculate it and it has the exact same place in the literature.
>
>
> **I found the double descent results in Sec 3.2 suggested issues with the proposed measure. Firstly, I don't see a double descent shape in the LHS plot. Then the right-hand plot seems (a) extremely noisy and (b) does not capture the correct U-shape that we would expect if it was working as we would hope. This seems to me to provide evidence that the measure is not working effectively.**
>
> The left hand side panel does show double descent. Usually in many papers, researchers do not use bootstrap to calculate the test error and that is why they see a distinct shape of the double descent. When you do bootstrap, there will be error bars on the double descent curve at all points and the curve will be more diffuse. The curve for MNSIT in Fig 3a uses bootstrap and has error bars, the curve for CIFAR-10 in Fig. 3b does not use bootstrap and you will see the familiar double descent plot.
>
> We have implemented these experiments carefully over almost two years, we do not believe there are mistakes here. For example, the training loss of models in our work is close to zero (0.01 - 0.1) in every case for deep learning models, so there are no under-fitting issues and that fits real practice well. The main spirit of the U-shape for deep learning is to reconcile the over-parameterization and generalization performance. Under our experimental condition that deep learning rarely underfit, we do show that higher effective dimension results in higher test loss.
>
> It is true that the right hand side plot does not show a U shape, we have discussed this in the paper. It is not noisy actually if you see it, the p-value for the slope being non-zero is significant in all cases except the one with N=50,000. This is very surprising in our opinion and should be investigated further. This does not mean that it is wrong. If the Reviewer can make a concrete argument otherwise, we will gladly consider it. The U shaped curve of MSE is a very formal argument and the details of this differ in different models.
>
> The brown line on the right panel which has a small slope corresponds to N=50,000. But it is actually very interesting that it does not show a linear trend. This shows that the learning capacity is almost constant as a function of the number of weights in the network (different brown points are different networks). This is in stark constant to the left panel where the test loss is non-monotonic in the number of weights of the network. In essence, when the number of samples available for training is very large, networks of the same architecture but different sizes have the same number of effective number of dimensions; this is presumably because there is not enough information in the dataset to constrain the large number of weights. A complement of this result is seen Fig 2 b (right) where networks of different architectures have different learning capacity at large sample sizes.

---

> ### Author Response · Authors · 2023-11-23
> **Response to Reviewer nwdU**
>
> **In Sec 3.3 and Fig 4, in contrast to the authors, I would find it hard to conclude that there is a particularly strong connection based on the results provided in the plot. I also find it hard to believe that the Pearson Correlation between the solid and dashed green lines is 0.99 based on visual inspection.**
>
> Sorry it should be Kendall-tau correlation, not Pearson. We have fixed this in the PDF now. The correlation is large because of the large error bars, and this why calculating the error bars is important.
>
> **No code was provided to inspect the experiments performed in more detail.**
>
> We will open-source the code with the camera ready. The code is pretty straight forward actually, it is training all the models as described in Section 2.3, fitting the expression for the learning capacity and plotting it.

---

### Official Review · Reviewer_anL3 · 2023-10-29

**Soundness:** 1 poor
**Presentation:** 1 poor
**Contribution:** 1 poor
**Rating:** 3
**Confidence:** 3

**Summary:**

The authors develop a theoretical analysis which begins with taking cross-entropy loss, and relating it to a probability distribution on the weights. They then use this distribution to compute the "energy" (average loss) and "learning capacity" (change in loss for the addition of one datapoint, times $N^2$). They then suggest that leave-one-out cross-validation estimator can be used to estimate this learning capacity.

The authors then develop a different cross-validation estimator to estimate the leave-one-out quantity, and use the method in combination with a sigmoid fit to try and estimate the learning capacity. They use this method to conduct experiments on MNIST, CIFAR10, WikiText-2, and a synthetic example to examine the links between the learning capacity and test accuracy. The analysis suggests that accuracy is inversely correlated with the learning capacity. They conclude with similar experiments on tabular datasets with non-deep learning methods.

**Strengths:**

The paper provides an interesting link between the probabilistic origins of losses like cross entropy, and the question of overparameterization/usefulness of data. Their numerical method for estimating the capacity provides an interesting measurement which is tractable for small and medium sized networks, and may lead to new insights.

**Weaknesses:**

Overall, I found the paper difficult to read, and some of the statements seem inaccurate. In addition, I found it hard to understand the link between the regime of validity of the theory and how it relates to the experiments.

One key difference between the thermodynamics partition function and this classifier partition function is that in thermodynamics, temperature and the number of terms are independent; in this analysis, the number of terms (data points) is exactly equal to the temperature. Indeed, in thermodynamics one takes N (or an equivalent quantity) to infinity, and the temperature emerges as a scale-independent constant. This fact should be present in the discussion of the analogy.

I think there's something wrong with the analysis of the learning capacity for quadratic energy; by my calculations, this quantity is related to the determinant of the matrix A (from the theory of multivariate Gaussian integrals) - not simply half the dimension $p/2$ as stated in the text. Indeed, in the same paragraph there is a formula given which depends on the spectrum of A, which is in contradiction to the estimate $p/2$ and the determinant form.

I also struggled to understand the link between the theoretical analysis, which involved a data-independent prior on the weight space and a dataset-induced probability distribution over weight space, and the experimental analysis, which involved looking at trained networks (that is, highly data-conditioned distributions on the weight space).

The analysis in Table 1 is far from convincing. On the real networks, the hypothesis doesn't hold for 1 out of the 3 examples. Each example itself consists of two datapoints. This does not give enough statistical power to resolve questions about the correlation between the capacity and the accuracy. In addition, the experiments showing that capacity doesn't show double descent behavior are not convincing as well; see questions.

Overall I don't think the authors have established the utility of their definition for capacity; the comparisons to other measures of capacity are a good start, but I think more careful experiments are needed to convincingly show that this is a quantity worth further study.

Maybe a better framing would involve just experimental results, empirically modeling the relationship between loss/accuracy and number of datapoints. As it currently stands, there is a big gap between the theory and experiments which makes the paper confusing to read and makes me doubt some of the conclusions.

**Questions:**

I'm a little confused about the steps taken in Equation 6; doesn't this directly follow from equation 1? Up to error in $O(N^{-2})$? I find the derivation in the text a bit hard to immediately grasp.

The analysis in section 2 seems to assume an abitrary prior on the parameters; I'm not sure how the fitting procedure plays into this, since that requires conditioning $w$ on the data. This left me with a lot of confusion on why the results from section 2 could be used in the setting of trained networks.

Why does the left panel on Figure 3 focus on the N  = 5e4 setting, when in the right hand panel it is the only curve that doesn't show a relationship between capacity and loss?

---

> ### Author Response · Authors · 2023-11-23
> **Response to Reviewer anL3**
>
> # ICLR 23 rebuttal
>
> ## **Reviewer anL3**
>
>
> Thank your for your feedback. We are very grateful to you for a careful reading of the manuscript and your questions. We believe we have addressed all your questions elaborately. There was some confusion regarding our constructions of the Bayesian posterior on the weight space and the calculations of the effective deimensionality. We have addressed them and also included an Appendix F that details the derivations. We hope you will consider increasing your score. Understanding generalization in deep learning is a difficult problem and papers such as these present a very fresh and unusual perspective upon the problem. They are valuable contributions to the literature. Our paper makes explicit connections to existing ideas in learning theory, which ensures that these attempts present a systematic research program to expand our collective body of knowledge.
>
> **Overall, I found the paper difficult to read, and some of the statements seem inaccurate. In addition, I found it hard to understand the link between the regime of validity of the theory and how it relates to the experiments.**
>
> Thank you for the response. Can you tell us which parts you don’t understand or which of the explanation is confusing to you? We are happy to elaborate here upon any part of the paper that you would like to explain better.
>
> **One key difference between the thermodynamics partition function and this classifier partition function is that in thermodynamics, temperature and the number of terms are independent; in this analysis, the number of terms (data points) is exactly equal to the temperature. Indeed, in thermodynamics one takes N (or an equivalent quantity) to infinity, and the temperature emerges as a scale-independent constant. This fact should be present in the discussion of the analogy.**
>
> This is a good point. The temperature in thermodynamics has an important meaning as a Lagrange multiplier of the entropy. The temperature in our problem setup does not mirror this construction. This is why we have merely stated it as an interesting analogy at the beginning of the paper. The most important part of the paper in this light is the connection of the learning capacity with the PAC-Bayes estimate of learning capacity in Section 2.3. Due to this, what seems like a pattern matching exercise between expressions of heat capacity in thermodyamics gets a rigorous grounding as an approximation of the PAC-Bayes effective dimensionality.
>
> **I think there's something wrong with the analysis of the learning capacity for quadratic energy; by my calculations, this quantity is related to the determinant of the matrix A (from the theory of multivariate Gaussian integrals) - not simply half the dimension ****as stated in the text. Indeed, in the same paragraph there is a formula given which depends on the spectrum of A, which is in contradiction to the estimate and the determinant form.**
>
> Sorry for the confusion. In the first bullet on Page 2, it should be “prior uniform prior”. We now provide calculation for the example mentioned in method (the convex energy one) in appendix F.
>
> **I also struggled to understand the link between the theoretical analysis, which involved a data-independent prior on the weight space and a dataset-induced probability distribution over weight space, and the experimental analysis, which involved looking at trained networks (that is, highly data-conditioned distributions on the weight space).**
>
> This is easy to answer. First, the Bayesian prior should always be data independent. Weights are sampled from this distribution and the training process induces a probability distribution over the weight space, i.e., the posterior, which is the Boltzmann distribution corresponding to Equation (2). This is a popular approach to thinking of the training process using the Bayesian perspective, e.g., in PAC-Bayes theory (McAllester, 1999) or variational analysis of SGD (Chaudhari & Soatto, 2018). The experiments use Monte Carlo to sample from this posterior. In practice, this means that we initialize the weights from the prior (a Gaussian in our case), train the network and take the trained network as a sample from the posterior---a pretty standard thing to do.

---

> ### Author Response · Authors · 2023-11-23
> **Response to Reviewer anL3**
>
> **The analysis in Table 1 is far from convincing. On the real networks, the hypothesis doesn't hold for 1 out of the 3 examples. Each example itself consists of two datapoints. This does not give enough statistical power to resolve questions about the correlation between the capacity and the accuracy. In addition, the experiments showing that capacity doesn't show double descent behavior are not convincing as well; see questions.**
>
> **Overall I don't think the authors have established the utility of their definition for capacity; the comparisons to other measures of capacity are a good start, but I think more careful experiments are needed to convincingly show that this is a quantity worth further study.**
>
> We are a bit puzzled by what the Reviewer is asking for here. Table 1 summarizes many experiments in this paper. For each configuration, e.g., a LeNet trained on MNIST, we calculated the learning capacity using many many bootstraps, cross-validation folds etc. and for this we also show that the estimate of the learning capacity correlates well with the test loss using the Kendall-tau coefficient in some cases. Across the hundreds of models trained on each configuration, for each sample size N, we have the test loss of trained model and an estimate of the learning capacity at that sample size. The Kendall-tau is calculated across these experiments. Each “example” that the Reviewer is referring to is totally independent experiment.
>
> The important point made in this table is that the learning capacity is a tiny fraction of the total number of weights. Please see similar tables in Jiang et al. (2019) which show similar (often weaker) correlations for all existing quantities that have been proposed as notions of capacity in deep learning. We have done experiments using many different models and many different datasets. We have made mathematical arguments that the learning capacity is closed related to PAC-Bayes bounds that are the best known bounds for understanding generalization in deep learning.
>
> This is not a speculative paper. We have provided plenty of evidence for our claims across many datasets and many architectures. The experiments in this paper (over the past ~1.5 years) were conducted using a lot of GPU time (many many Monte Carlo runs and MCMC runs until we were able to estimate these quantities robustly). Altogether, the data presented in the paper uses results from ~30,000 neural networks of varying complexities trained on MNIST and CIFAR-10, ~5,400 random forests (each with 1000 trees), and optimizing the PAC-Bayes bound ~2,800 times (which is very expensive computationally). This is a lot of experiments.
>
> The experiments on double descent are actually extremely exhaustive. Please note that the current double descent plot on MNIST in Fig 4 was obtained by training on 14 different values of N (number of samples), and 4 bootstrapped datasets for each value of N, 5 cross-validation-style folds, and 5 models on each fold. This amounts to 1400 models on MNIST for a single architecture. There are 6 architectures, which amounts to 8,400 models in total. The experiments in this paper are already quite exhaustive (and already quite expensive). We have also included a similar plot on CIFAR-10 in the updated manuscript; this roughly 25x times more expensive computationally to create. The results of the double descent plot are actually extremely impressive. This is because to date, there is no other quantity that allows us to eliminate double descent---this one does.
>
> **Maybe a better framing would involve just experimental results, empirically modeling the relationship between loss/accuracy and number of datapoints. As it currently stands, there is a big gap between the theory and experiments which makes the paper confusing to read and makes me doubt some of the conclusions.**
>
> We respectfully disagree. Our attempts at understanding generalization in deep learning should be systematic and constructive. Simply proposing a quantity and checking whether it relates to generalization is a difficult proposition because there are a lot of bells and whistles in deep networks and different situations might give us very different conclusions. The utility of doing things from first principles, and connecting such ideas with existing ideas in the field, is that we know precisely the settings where the ideas will work, and how we should test them in practice.
>
> The gap between theory and practice that  you talk about in this paper is actually a reflection of the large gap in between the two in our field. Deep networks work extremely well in practice and yet our mathematical understanding of why they do so well is poor---this poor. Our paper is an attempt at crossing this bridge. And papers like this are important stepping stones on the path to a broader theory that will emerge in the community.

---

> > ### Author Response · Authors · 2023-11-23
> > **Response to Reviewer anL3**
> >
> > **I'm a little confused about the steps taken in Equation 6; doesn't this directly follow from equation 1? Up to error in O( N − 2) ? I find the derivation in the text a bit hard to immediately grasp.**
> >
> > We now provide a more detailed derivation in Appendix F.
> >
> > **The analysis in section 2 seems to assume an abitrary prior on the parameters; I'm not sure how the fitting procedure plays into this, since that requires conditioning w on the data. This left me with a lot of confusion on why the results from section 2 could be used in the setting of trained networks.**
> >
> > There seems to be a misunderstanding here. **** Please see our response to your question above on “I also struggled to understand the link between the theoretical analysis, which involved a data-independent prior on the weight space and a dataset-induced probability distribution over weight space....”
> >
> > **Why does the left panel on Figure 3 focus on the N = 5e4 setting, when in the right hand panel it is the only curve that doesn't show a relationship between capacity and loss?**
> >
> > The number N = 50,000 is the largest number of samples available in the train set of MNSIT. We computed the double descent for N=50,000 on the left panel only because that is what is often done in the literature. This panel was obtained by training on 14 different values of N (number of samples), and 4 bootstrapped datasets for each value of N, 5 cross-validation-style folds, and 5 models on each fold. This amounts to 1400 models on MNIST for a single architecture. There are 6 architectures, which amounts to 8,400 models in total. So this is an expensive experiment to run even for one sample size.
> >
> > The brown line on the right panel which has a small slope corresponds to N=50,000. But it is actually very interesting that it does not show a linear trend. This shows that the learning capacity is almost constant as a function of the number of weights in the network (different brown points are different networks). This is in stark constant to the left panel where the test loss is non-monotonic in the number of weights of the network. In essence, when the number of samples available for training is very large, networks of the same architecture but different sizes have the same number of effective number of dimensions; this is presumably because there is not enough information in the dataset to constrain the large number of weights. A complement of this result is seen Fig 2 b (right) where networks of different architectures have different learning capacity at large sample sizes.

---

### Official Review · Reviewer_4vaU · 2023-11-02

**Soundness:** 3 good
**Presentation:** 2 fair
**Contribution:** 2 fair
**Rating:** 3
**Confidence:** 3

**Summary:**

This work introduces a Monte Carlo-based procedure to estimate the learning capacity of machine learning models. Learning capacity is a learning theoretical counterpart of heat capacity and quantifies the local change of test loss (averaged over a distribution of weights) when increasing the size of the training data. It is also a measure of (effective) model dimension. Empirically, it is shown that the estimate of learning capacity correlates with testing error and can thus be used for model selection. Moreover, a connection to the effective dimension as defined in Yang et al., ICML 2022 is derived.

**Strengths:**

- Learning capacity can be computed for a wide range of learning models, including neural networks and non-parametric models such as k-nearest neighbor and random forest.
- Learning capacity correlates well with test loss.

**Weaknesses:**

1. Learning capacity is a measure that is computed on the testing data as a function of the test loss. Specifically, it is the derivative of the population loss (averaged over a distribution of trained models) with respect to training set size. Thus, it is unsurprising, that this quantity correlates well with test loss.
Moreover, it is costly to compute, as multiple models need to be trained (to sample from the model distribution) for differing sizes of training data (to quantify the change of test loss with training set size). Overall, as learning capacity requires access to the testing split, it appears to be a complicated and inefficient replacement of a validation loss.

2. It is argued that the estimator of learning capacity can be used to calculate a PAC-Bayes bound. (*"Our techniques to estimate the learning capacity are computationally inexpensive ways to calculate PAC-Bayes bounds.*") This is not true. Such a bound must account for the uncertainty in the estimate of learning capacity, however, this work does not present bounds on the quality of the estimator.

3. The arguments are difficult to follow. Among others, I found the presentation of the connection between learning capacity and PAC-Bayes theory in section 2.3 rather confusing.

4. When reading this paper I got the impression, that the concept of learning capacity is novel and part of the contribution. This is not the case, as learning capacity and its properties (e.g. as a measure of dimensionality) can already be found in LaMont & Wiggins (2019). The latter is cited, but a comparison is only given in the supplementary material. Instead, it should have been placed prominently in the main part, to avoid giving a wrong impression.

5. The Monte-Carlo based estimator for learning capacity, which is listed as a main contribution, is only presented in the supplementary material.

**Questions:**

*"Given these estimates of the average energy, we could now calculate $C = −N^2 \partial_N U$ but it is difficult to do so because the noise in the estimation of U gets amplified when we take the finite-difference derivative."*
Could you clarify, whether this is an observation or a mathematical fact?

Which value did you set $\epsilon$ parameter for computing the effective dimensionality from Yang et al. in section 3.3 to (Yang et al. state that the parameter has to be user chosen)? How did you select it?

---

> ### Author Response · Authors · 2023-11-23
> **Response to Reviewer 4vaU**
>
> # ICLR 23 rebuttal
>
>
> ## **Reviewer 4vaU**
>
> Thank you for your feedback on our paper. We have addressed your comments below. This is a theoretical paper and makes the point that the learning capacity can be used to understand the remarkable generalization of deep learning. We are neither proposing that this the only such quantity, nor are we saying that this is the complete story of generalization in deep learning. But our paper does provide sufficient evidence that this quantity is meaningful because it relates well to test loss, interesting in the sense that lets us study and discover new phenomena (e.g., a very small effective dimensionality and freezing), and closely related to existing theories of generalization in deep learning (e.g., PAC-Bayes bounds). It is therefore a worthwhile addition to the literature.
>
> **Learning capacity is a measure that is computed on the testing data as a function of the test loss. Specifically, it is the derivative of the population loss (averaged over a distribution of trained models) with respect to training set size. Thus, it is unsurprising, that this quantity correlates well with test loss.**
>
> The population loss is an average over the data generating distribution. How can it be unsurprising that the derivative of the empirical estimate of population loss with respect to the number of samples correlates with the population loss. The derivative of a function is very different from its value. Furthermore, our paper also makes the point that the learning capacity is a very small fraction of the number of weights. These are non-trivial observations. If the Reviewer has an alternative calculation that could explain the observations in our paper, we would be very grateful if you can point them to us.
>
> There is further point we would like to emphasize. There are many quantities in learning theory like VC dimension, the right hand-side of PAC-Bayes bounds, etc. All of these quantities are by definition related to the test error. But there is an intense discussion in the literature as to how we can accurately estimate these quantities. Our paper studies another quantity of this kind, the learning capacity. We develop methods to estimate it. And the fact that it correlates well with the test error shows that both the quantity and our methods to estimate it are a useful contribution to the literature.
>
> **Moreover, it is costly to compute, as multiple models need to be trained (to sample from the model distribution) for differing sizes of training data (to quantify the change of test loss with training set size). Overall, as learning capacity requires access to the testing split, it appears to be a complicated and inefficient replacement of a validation loss.**
>
> It depends a bit on your point of view. This is a theoretical paper. Our goal is not to propose a quantity that replaces the validation loss. Indeed, the validation loss is a consistent and pretty inexpensive way to estimate the test loss. So one would always use it over any quantity like VC dimension, or PAC-Bayes bounds, in a real application (this point was also made in https://arxiv.org/abs/1710.05468). But the reason we study these quantities in the literature is that we they help us characterize the principles that govern generalization.
>
> The big issue in deep learning is that these quantities, even if they do tell us these principles, do not seem to correlate well with the test loss. The learning capacity does correlate with the test loss, so it mitigates this issue. The second issue is that some of the standard learning theoretic quantities are difficult to estimate for deep networks, e.g., PAC-Bayes bounds work well but it is very expensive to calculate them. The learning capacity is an approximation of PAC-Bayes-based effective dimensionality (Equation 13) and easier to calculate.
>
> Even if estimating the learning capacity is expensive, we glen a lot of interesting insights from doing so. For example, we see that the number of degrees of freedom constrained by the data is much fewer than the number of weights. This insight is not seen using estimates of other quantities in learning theory. It also suggests that we should focus future theoretical analysis on understanding why the data is so “simple”; this seems to be at heart of why deep networks are working well. Phenomena such as double descent which have been widely studied and talked about in the literature can be understood in a new light using the learning capacity. It takes a bit more computer time to glean these insights but that is a cost well paid for developing a better understanding of deep networks because it is such an important and broadly impactful problem.

---

> > ### Author Response · Authors · 2023-11-23
> > **Reviewer 4vaU**
> >
> > **It is argued that the estimator of learning capacity can be used to calculate a PAC-Bayes bound. ("Our techniques to estimate the learning capacity are computationally inexpensive ways to calculate PAC-Bayes bounds.") This is not true. Such a bound must account for the uncertainty in the estimate of learning capacity, however, this work does not present bounds on the quality of the estimator.**
> >
> > Sorry for the confusion. In Section 2.3, the PAC-Bayes bound is presented, encompassing a term that approximates the learning capacity. In our analysis, minimizing the expression introduced in Section 2.3 can be interpreted as a trade-off between the distance separating the posterior and prior distributions, model complexity, and the training loss.
> >
> >
> > **The arguments are difficult to follow. Among others, I found the presentation of the connection between learning capacity and PAC-Bayes theory in section 2.3 rather confusing.**
> >
> > The presentation in Section 2.3 is a bit difficult to follow because it is a compilation of many results in the PAC-Bayes literature. Specifically, we are discussing the techniques used by the paper by Yang et al. (2022b) to define a kind of effective dimensionality (Equation 13). That paper gives rigorous and elaborate derivations of these formulae; it also showed that PAC-Bayes bounds can give a non-vacuous analytical expression for the upper bound on the generalization error. In Section 2.3, we are making the argument that their expression for the effective dimensionality is close to our definition for the learning capacity (Equation 15). This connection with PAC-Bayes also gives learning capacity a firm theoretical foundation.
> >
> > **When reading this paper I got the impression, that the concept of learning capacity is novel and part of the contribution. This is not the case, as learning capacity and its properties (e.g. as a measure of dimensionality) can already be found in LaMont & Wiggins (2019). The latter is cited, but a comparison is only given in the supplementary material. Instead, it should have been placed prominently in the main part, to avoid giving a wrong impression.**
> > ** **
> >
> > We have cited Lamont and Wiggins’ paper on Page 2 itself. We provide a response to this question in the Appendix G. We repeat it here. We will add an additional sentence that cites the paper again after Equation (4) to make it clear to the reader that these quantities were also studied by Lamont and Wiggins’ paper.
> >
> > Our paper is strongly inspired by the paper of Lamont and Wiggins. They coined the term “learning capacity” and showed that for some systems (a free particle in vacuum and Gaussian distributions), the learning capacity is proportional to the number of degrees of freedom. They suggested that one could calculate U using finite differences of the log-partition function (note, estimating its derivative which is the learning capacity C robustly is the difficult step in this approach).
> >
> > We used the learning capacity to investigate the generalization of machine learning models. All our findings from Section 2.2, especially all the insights discussed in the ”Results” section are new. To glean these insights, we made the analogy of Lamont and Wiggins more rigorous, by drawing out a clean connection PAC-Bayes theory (this is critical because it shows that what is essentially a pattern matching exercise is meaningful) and to singular learning theory (in their paper, it was more of a speculation). We have developed an effective procedure to numerically calculate the learning capacity (which is absent in the original paper).
> >
> > We should emphasize that heat capacity is a very basic thermodynamic quantity. Learning capacity is completely analogous, except that we interpret the statistical ensemble in physics as all possible models. The ideas from thermodynamics used in this paper are in standard undergraduate textbooks on the subject. In particular there is nothing new that we say up to Section 2.1. The correspondence between the number of samples and the inverse temperature is simply because the factor of 1/N in (1) shows up in the exponent in (2).
> >
> >
> > **The Monte-Carlo based estimator for learning capacity, which is listed as a main contribution, is only presented in the supplementary material.**
> >
> >
> > We had to make some editorial choices to fit the material within the 9 pages.

---

> > > ### Author Response · Authors · 2023-11-23
> > > **Reviewer 4vaU**
> > >
> > > **Given these estimates of the average energy, we could now calculate C = − N 2∂N U but it is difficult to do so because the noise in the estimation of U gets amplified when we take the finite-difference derivative." Could you clarify, whether this is an observation or a mathematical fact?**
> > >
> > > This is an empirical observation. But you can appreciate why it could be true by looking at the expression in Equation (6,7,8).
> > >
> > > **Which value did you set ε parameter for computing the effective dimensionality from Yang et al. in section 3.3 to (Yang et al. state that the parameter has to be user chosen)? How did you select it?**
> > >
> > > We optimized the PAC-Bayes bound using the methods/code of Yang et al. to select the values of epsilon. We will make a table in the Appendix that details these values. Optimizing PAC-Bayes bounds correctly is tricky but we have a lot of experience in doing so. And as the numerical estimates suggest, our results are also consistent with existing results in the literature.

---

### Official Review · Reviewer_mHxX · 2023-11-02

**Soundness:** 2 fair
**Presentation:** 3 good
**Contribution:** 4 excellent
**Rating:** 6
**Confidence:** 4

**Summary:**

Motivated by concepts from statistical mechanics in thermodynamics and the work of (LaMont and Wiggins, 2019), the authors propose the heat capacity (here called the learning capacity), defined as the scaled second derivative of the tempered log-marginal likelihood (or free energy), as a measure of intrinsic model dimension. It is shown empirically that learning capacity often anticorrelates with test accuracy, and that the ratio of the learning capacity to the number of model parameters is often small. A variety of smaller models are considered with a variety of standard (again, smaller) datasets. The test loss is also shown to not exhibit double descent when plotted as a function of learning capacity instead of number of model parameters. Theoretical PAC-Bayes-like arguments are presented to support a connection between the learning capacity and the test error. Behaviour of the learning capacity is examined for small and large sample sizes, showing that in either case, the learning capacity plateaus to common values, indicating diminishing returns. To estimate the learning capacity, its behaviour with respect to the sample size is parameterised using a sigmoid function. This expression is integrated to get a parameterised form of the average energy. Since the average energy can be estimated using leave-one-out cross-validation, the parameters of the sigmoid model for the learning capacity can be estimated using nonlinear least-squares fits. Another approach using MCMC is also presented. These methods do not rely on any parameterisation, so the learning capacity for non-parametric models is also investigated. The appendices also show that, asymptotically in the sample size, the learning capacity is equal to the real-log canonical threshold (RLCT) from singular learning theory.

**Strengths:**

- Based on a very general theory of statistical learning.
- Theoretical arguments are (mostly) well-explained.
- Connections to the PAC-Bayes setup are compelling, as this framework has been shown to reveal the tightest estimates of generalisation performance to date.
- Two algorithms are proposed, with neither requiring the prior, and one not even requiring parameterisation!
- A variety of novel, cool experiments: monotonic relationships between test loss, learning capacity, and PAC-Bayes effective dimensionality are compelling.
- Interesting connection with singular learning theory.
- Experiments are conducted on a fairly wide range of models.

**Weaknesses:**

- Derivation of the PAC-Bayes bound could be much more rigorous than presented. The assumptions should be made clear, and the full bound provided as a theorem.
- Theoretical arguments all rely on a non-singular Hessian, which is never the case for practical neural networks. See, for example [1].
- Models considered are relatively small, and not nearly state-of-the-art. This can be forgiven, due to the complexities associated with estimating Bayesian quantities, but is nonetheless an issue that would need to be overcome eventually for the work to be practical.
- While the approximation methods for the learning capacity appear to be effective, they also appear to be very expensive, requiring multiple trained models over bootstrapped versions of the datasets, or a lengthy MCMC chain.

[1] Wei, S., Murfet, D., Gong, M., Li, H., Gell-Redman, J., & Quella, T. (2022). Deep learning is singular, and That’s good. IEEE Transactions on Neural Networks and Learning Systems.

**Questions:**

- Can you write a full PAC-Bayes bound involving the learning capacity under some approximating assumptions? This, together with the first listed "weakness", is probably my biggest gripe with the paper, and I would be happy to improve my score if the authors can show this is achievable.
- By linearising the model akin to NTK, can the Hessian be replaced by the empirical NTK? All the following arguments would still hold, but now the eigenvalues would be for the empirical NTK instead, and would therefore be well-defined.  Similar ideas are used in [2].
- The section headings don't seem like the ICLR defaults, and they are a little unpleasant. Can you change them back to the defaults?
- How long does it take to estimate the learning capacity for a single model? Can you report this in terms of how much longer it takes than training a single model?
- Do the numerical procedures (particularly the MCMC one) yield consistent estimators of the learning capacity? I'm guessing not, if the sigmoid parameterisation is necessary in both cases.

[2] Hodgkinson, L., van der Heide, C., Salomone, R., Roosta, F., & Mahoney, M. W. (2023). The Interpolating Information Criterion for Overparameterized Models.

**Details Of Ethics Concerns:**

None.

---

> ### Author Response · Authors · 2023-11-23
> **Response to Reviewer mHxX**
>
> # ICLR 23 rebuttal
>
>
>
> ### **Reviewer mHxX**
>
> Thank you for your careful reading of the manuscript and the detailed feedback. Thank you also for your willingness to increase your score. We believe have addressed all our questions below. We you will consider championing our paper. It is unlikely that a single paper will finish the story on understanding the impressive generalization of deep networks. But in our opinion the merit of papers like ours is that they give useful pointers for developing future theory. Now that we know that the number of constrained degrees of freedom in a network is much smaller than the number of weights and that could explain the generalization in deep learning, we should next aim to understand why it is so.
>
> **Derivation of the PAC-Bayes bound could be much more rigorous than presented. The assumptions should be made clear, and the full bound provided as a theorem.**
>
> Thank you. We have expanded the Appendix to include the derivations of the different expressions of the learning capacity, the log partition function and the PAC-Bayes bound.
>
> **Theoretical arguments all rely on a non-singular Hessian, which is never the case for practical neural networks. See, for example [1].**
>
> Thank you for pointing us to this paper. We will add a discussion to the updated manuscript. There are indeed direct connections between our calculations and singular learning theory. We provided a majority of our derivations using non-singular Hessians because this is easy. But we also discuss singular Hessians, e.g., in the narrative on Page 3 before Equation (5). Appendix C also details these connections precisely; in an initial submission of our paper we found that the readers were confused with the exposition of singular learning theory (which is rather uncommon in mainstream machine learning literature...this is the reason we moved that content to the Appendix.
>
> As the Reviewer might realize, the merit of our calculations is that the learning capacity gives the correct expressions for the effective dimensionality even for singular models. And this is also evident in our experiments with deep networks, which are singular almost everywhere.
>
> **Models considered are relatively small, and not nearly state-of-the-art. This can be forgiven, due to the complexities associated with estimating Bayesian quantities, but is nonetheless an issue that would need to be overcome eventually for the work to be practical.**
>
> Fair point. As we discuss in the Introduction, our perspective on this is as follows. PAC-Bayes bounds correlate very well with the test error of deep networks. But beyond small multi-layer preceptors on MNIST of CIFAR-10, it is very difficult to calculate PAC-Bayes bounds without using compression. We see our work as approximating the PAC-Bayes calculations using Monte Carlo and this allows us to scale to much larger networks.
>
> To wit, the networks considered here may be small by the standards of foundation models, but they are not really all that small. The LSTM on WikiText-2 has ~14M weights, and the WideResnet on CIFAR-10 achieves a test accurate of 93.7%  which is very good. These techniques can be implemented as is for large networks. The MCMC procedure in Appendix B gives an even more efficient mechanism to estimate the learning capacity. Altogether, we believe that these techniques provide evidence that deep networks have much fewer effective degrees of freedom than what the number of weights indicate and that when we consider the test loss as a function of these effective degrees of freedom, we do not see double decent. These are the main points of our paper.
>
> Indeed, research on Bayesian statistic for large models will require high computational complexity but we regard our work as a start. The calculations and experimental results give intuitive understanding and effectiveness of the quantity. By proving that this quantity can be a correct measure, the result can be used as a foundation to justify the soundness of a theory or can be used as a quantity to check whether a model saturate or not in terms of performance. We leave the further improvement as future work.

---

> ### Author Response · Authors · 2023-11-23
> **Response to Reviewer mHxX**
>
> **While the approximation methods for the learning capacity appear to be effective, they also appear to be very expensive, requiring multiple trained models over bootstrapped versions of the datasets, or a lengthy MCMC chain.**
>
> Yes, quantity involving Bayesian statistic often require boostrapped or sampling from a posterior given an algorithm. For this paper, since our goal is to make a convincing argument for the utility of the learning capacity, we have focused on accurate estimation using a lot of Monte Carlo. For practical applications, one could implement fewer bootstraps, or using the MCMC method described in the Appendix. This may not be accurate but it will likely be good enough for decision making, e.g., when a model is data starved, vs. when there is sufficient amount of data and we should focus on the architecture; this is an important practical implication of the freezing phenomenon discussed in our paper.
>
> **Can you write a full PAC-Bayes bound involving the learning capacity under some approximating assumptions? This, together with the first listed "weakness", is probably my biggest gripe with the paper, and I would be happy to improve my score if the authors can show this is achievable.**
>
> Thank you for your willingness to increase your score. We have provided detailed derivation of the PAC-Bayes bounds in Appendix F now.
>
> **By linearising the model akin to NTK, can the Hessian be replaced by the empirical NTK? All the following arguments would still hold, but now the eigenvalues would be for the empirical NTK instead, and would therefore be well-defined. Similar ideas are used in [2].**
>
> This is a very good idea. We believe it will work well. In other work, we have also calculated the Gauss-Newton approximation of the Hessian, which is equal to the empirical Fisher information matrix. The eigenvalues of such matrices can be calculated using a Krocnker factorization of the blocks corresponding to each layer; so it can be implemented quite efficiently. The bigger issue with estimating the effective dimensionality using the Hessian/NTK etc. is however that the scale of the prior epsilon in Equation (5) needs to be calculated by minimizing the right-hand side of the PAC-Bayes bounds. Without a good estimate of epsilon, one cannot get a useful estimate of C. This is the difficult part in our experience and that is why we have developed the Monte Carlo procedure to directly estimate C.
>
> **The section headings don't seem like the ICLR defaults, and they are a little unpleasant. Can you change them back to the defaults?**
>
> Sure, we will do this.
>
> **How long does it take to estimate the learning capacity for a single model? Can you report this in terms of how much longer it takes than training a single model?**
>
> The time required to estimate the learning capacity depends upon the number of bootstraps and the number of models with random initial conditions. For our experiments, we used 14 different sample sizes, 4 bootstrapped datasets per sample size and 5-fold cross-validation with 5 models for each fold. Training with small sample sizes is less expensive than large ones. So roughly, we need time that is roughly equal to (14/2)*4*25 =  700 times the time for training one model on the entire dataset. This may seem a lot but it can be reduced quickly, if necessary, in a real application.
>
> **Do the numerical procedures (particularly the MCMC one) yield consistent estimators of the learning capacity? I'm guessing not, if the sigmoid parameterisation is necessary in both cases.**
>
> The two are quite close. We have experimented with and without the sigmoid parameterization for many months. The estimates of learning capacity without the sigmoid parameterization are close to the ones presented here but the error bars are larger; this is the same for the MCMC procedure... the error bars are larger. We therefore started using the sigmoid-based parameterization. We will add the results without sigmoid parameterization in the Appendix for the updated manuscript.

---

### Meta-Review · Area_Chair_2Fsx · 2023-12-08

**Metareview:**

The authors present a theory of model capacity inspired by thermodynamics which illuminates some of the paradoxical aspects of double descent, as well as supposedly having close connections to PAC-Bayes and Singular Learning Theory. While all of the reviewers found the ideas interesting, most had objections to the style of the paper, which they often found hard to follow or lacking in rigor. Some reviewers also objected to the small scale of the experiments, but for a paper on theory this is acceptable and not a reason for rejection in my opinion. Nevertheless, given the concerns around the presentation, it does not sound like the paper is ready to be accepted at this time. It may be that the presentation is simply too dense for a standard conference paper, in which case I would recommend the authors either expand the paper and submit it to a venue like TMLR or JMLR, or split the paper into a series of papers on more focused aspects of the work.

As a side note, I am disappointed that most of the reviewers chose not to engage with the authors despite the significant effort the authors put into their rebuttal. However, since none of the reviews which recommended rejection were borderline, I do not believe that more reviewer engagement would have significantly changed the outcome.

**Justification For Why Not Higher Score:**

This was addressed in the metareview.

**Justification For Why Not Lower Score:**

This was addressed in the metareview.

---

### Decision · Program_Chairs · 2024-01-16

Reject